# Faster Reinforcement Learning with Value Target Lower Bounding

## Abstract

We show that an arbitrary lower bound of the maximum achievable value can be used to improve the Bellman value target during value learning. In the tabular case, value learning using the lower bounded Bellman operator converges to the same optimal value as using the original Bellman operator, at a potentially faster speed. In practice, discounted episodic return in episodic tasks and n-step bootstrapped return in continuing tasks can serve as lower bounds to improve the value target. We experiment on Atari games, FetchEnv tasks and a challenging physically simulated car push and reach task. We see large gains in sample efficiency as well as converged performance over common baselines such as TD3, SAC and Hindsight Experience Replay (HER) in most tasks, and observe a reliable and competitive performance against the stronger n-step methods such as td-lambda, Retrace and optimality tightening. Prior works have already successfully applied a special case of lower bounding (using episodic return), but are limited to a small number of episodic tasks. To the best of our knowledge, we are the first to propose the general method of value target lower bounding (with possibly bootstrapped return), to demonstrate its optimality in theory, and effectiveness in a wide range of tasks over many strong baselines.

## 1 Introduction

The value function is a key concept in dynamic programming approaches to Reinforcement Learning (RL) (Bellman, 1957). It estimates the sum of all future rewards (usually time-discounted) of a given state. In temporal difference (TD) learning, the value function is adjusted toward its Bellman target which adds the reward of the current step with the (discounted) value of the next state (Sutton & Barto, 2018). This forms the basis of many state of the art RL algorithms such as DQN (Mnih et al., 2013), DDPG (Lillicrap et al., 2016), TD3 (Fujimoto et al., 2018), and SAC (Haarnoja et al., 2018).

The value of the next state is typically estimated using a "bootstrapped value" based on the value function itself, which is being actively learned during training. The bootstrapped values can be random and far from the optimal value, especially at the initial stage of training, or with sparse reward tasks where rewards can only be achieved through a long sequence of actions. Consequently, the Bellman value targets as well as the learned values are usually far away from the optimal value (the value of the optimal policy).

Naturally, this leads to the following idea: If we can make the value target closer to the optimal value, we may speedup TD learning. For example, we know that the optimal value is just the expected discounted return of the optimal policy, which always upper bounds the expected return of any policy. For episodic RL tasks, we could use the observed discounted return up to episode end from the training trajectories to lower bound the value target. This makes the new value target closer to the optimal value, when the empirical return is higher than the Bellman target.

Submitted to 36th Conference on Neural Information Processing Systems (NeurIPS 2022). Do not distribute.

---
**Algorithm 1** Value iteration with value target lower bounding
---
**Input:** Finite MDP $p(s', r|s, a)$, convergence threshold $\theta$, a lower bound $f(s)$ of the maximum achievable value $\bar{G}^v(s)$
**Output:** State value $v(s)$
$v(s) \leftarrow 0$
**repeat**
  $\Delta \leftarrow 0$
  $v_p(s) \leftarrow v(s)$
  **for** each state $s$ **do**
    $\hat{v}(s) \leftarrow \max_a \sum_{s',r} p(s', r|s, a)[r + \gamma v_p(s')]$
    $\hat{v}_f(s) \leftarrow \max(f(s), \hat{v}(s))$
    $v(s) \leftarrow \hat{v}_f(s)$
    $\Delta \leftarrow \max(\Delta, |v(s) - v_p(s)|)$
  **end for**
**until** $\Delta < \theta$
---

The case for continuing or non-episodic tasks is less clear though. When a continuing task can return negative rewards, any safe lower bound of the optimal value can be too low to be useful. One could take the risk and use n-step bootstrapped return as a lower bound, which is unsafe because bootstrapped return can overestimate and be greater than the optimal value. Can we still use them as lower bounds to improve TD value targets?

## 2 Theoretical Results for the Tabular Case

Our results show that for the tabular case, arbitrary functions below a certain bootstrap bound can be used to lower bound the value target to still converge to the same optimal value.

### 2.1 Background

In finite MDPs with a limited number of states and actions, a table can keep track of the value of each state. Using dynamic programming algorithms such as value iteration, values are guaranteed to converge to the optimum through Bellman updates (Chapter 4.4 (Sutton & Barto, 2018)).

The core of the value iteration algorithm (Algorithm 1) is the Bellman update of the value function, $\mathcal{B}(v)$, where $v(s')$ is the bootstrapped value:

$$\mathcal{B}(v)(s) := \max_a \sum_{s',r} p(s', r|s, a)[r + \gamma v(s')] \tag{1}$$

It is well known that the Bellman operator, $\mathcal{B}$, is a contraction mapping over value functions (Denardo, 1967). That is, for any two value functions $v_1$ and $v_2$, $||\mathcal{B}(v_1) - \mathcal{B}(v_2)||_\infty \leq \gamma ||v_1 - v_2||_\infty$ for the discount factor $\gamma \in [0, 1)$ and $||x||_\infty := \max_i |x_i|$ (the $L_\infty$ norm). This guarantees that any value function under the algorithm converges to the optimal value $\mathcal{B}^\infty(v) = v^*$.[1]

### 2.2 Convergence of value target lower bounding

**Definition 2.1.** The expected n-step bootstrapped return for a given policy $\pi$ and value function $v(s)$ is defined as the expected bootstrapped return of taking $n$ steps according to policy $\pi$:

$$G_n^{\pi,v}(s_0) := \mathbb{E}^\pi \{r_1 + ... + \gamma^{n-1} r_n + \gamma^n v(s_n)\} \tag{2}$$

Here, the step rewards $r_i$ and the resulting n-th step state $s_n$ are random variables, with the expectation $\mathbb{E}^\pi$ taken over all possible n-step trajectories under the policy $\pi$ and the given MDP.

---

[1]For the gist of the proof, see for example page 8 of `https://people.eecs.berkeley.edu/~pabbeel/cs287-fa09/lecture-notes/lecture5-2pp.pdf`

**Definition 2.2.** Given the current learned value function $v(s)$, policy class $\Pi$, the *maximum achievable value* of a state $s$ is defined as:

$$\bar{G}^v(s) := \max_{\pi \in \Pi, n \in [1, +\infty)} G_n^{\pi, v}(s) \tag{3}$$

This is a more relaxed definition of maximum because for each state $s$, a different policy $\pi(s)$ and a different number of steps $n(s)$ can be used to achieve the maximum $\bar{G}^v(s)$. And the theorem below says any function not exceeding the maximum achievable value can be used to lower bound the value target, and still achieve the optimal value in convergence.

**Theorem 2.3.** *Under the same assumptions for Bellman value contraction, for any function $f$ that lower bounds the maximum achievable value, i.e. $\forall s, f(s) \leq \bar{G}^v(s)$, if we define the lower bounded Bellman operator as $\mathcal{B}_f(v) := \max(\mathcal{B}(v), f)$, then $\mathcal{B}_f^\infty(v) = \mathcal{B}^\infty(v)$.*

Note, the value $v(s)$ and the bootstrapped value can be inaccurate, and even above the optimal value. As a consequence, when $n$ is finite, the maximum achievable value $\bar{G}^v(s)$ (and $f$) can be above the maximum expected return (i.e. the optimal value). On the other hand, when $n$ is sufficiently large, the effect of the bootstrap value $v(s_n)$ diminishes (see Equation 2), and the maximum achievable value becomes the maximum expected return (i.e. the optimal value). Therefore, $\forall s, \bar{G}^v(s)$ is no smaller than the optimal value $\mathcal{B}^\infty(v)(s)$. As a special case of the theorem, as long as $f$ is below the optimal value, value target lower bounding converges correctly:

**Corollary 2.4.** *If function $f$ lower bounds the optimal value, i.e. $\forall s, f(s) \leq \mathcal{B}^\infty(v)(s)$, then $\mathcal{B}_f^\infty(v) = \mathcal{B}^\infty(v)$.*

A few things to note about the proof of Theorem 2.3 (included in Appendix 1.1).

First, this only proves convergence, not contraction under the original $||v_1 - v_2||_\infty$ metric. In the case of the Bellman operator, contraction shows that $\forall v_1, v_2$ value functions, $||\mathcal{B}(v_1) - \mathcal{B}(v_2)||_\infty \leq \gamma ||v_1 - v_2||_\infty$. Here, for value target lower bounding, what's proved is convergence to $v^*$ at a rate of $\gamma$, not contraction. There can be counter examples where the distance between $v_1$ and $v_2$ under one application of $\mathcal{B}_f$ can increase in the original $L_\infty$ metric space, even though $v_1$ and $v_2$ are both getting closer to $v^*$ at a rate of $\gamma$. One difficulty caused by convergence instead of contraction is that the stopping criterion in Algorithm 1 ($\Delta < \theta$) no longer works, due to the inaccessible $v^*$ during learning. In practice, this may not be a serious concern, as people often train algorithms for a fixed number of iterations or time steps.

Second, based on the proof, the new algorithm is at least as fast as the original. When the lower bound actually improves the value target, i.e. $f(s) > \mathcal{B}(v_1)(s)$, there is a chance for the convergence to be faster. Convergence is strictly faster when the lower bound $f$ has an impact on the $L_\infty$ distance between the current value and the optimal value, i.e. it increases the value target for the states where the differences between the current value and the optimal value are the largest.

Third, the lower bound function doesn't have to be static during training. As long as there is a single $f$ during each training update, convergence is preserved.

The following sections detail how to compute lower bounds of the maximum achievable value (Section 3), how to integrate the lower bounds into state of the art RL algorithms (Section 4), and provide an illustration of how this method may benefit value learning in practice (Section 4.3).

# 3 Example Lower Bound Functions

We show a few cases where lower bound functions can be readily obtained from the training experience. Future work may investigate alternatives.

## 3.1 Episodic tasks

In episodic tasks, discounted return is accumulated up to the last step of an episode. In this case, we can wait until an episode ends, and compute future discounted returns of all time steps up to the end of the episode. This episodic return is a lower bound of the optimal value when the environment is

deterministic, because the reward sequence can be repeated using the same sequence of actions[2]. To make training efficient, we can compute and store such discounted returns into the replay buffer for each time step, and simply read them out during training, which adds very little computation to the baseline one-step TD computation.

$$f(s_0) = \sum_{i=0,..,\infty} \gamma^i r(s_i, a_i) \tag{4}$$

We call this variant "lb-DR", short for lower bounding with discounted return.

### 3.1.1 Episodic with hindsight relabeled goals

In goal conditioned tasks, one helpful technique is hindsight goal relabeling (Andrychowicz et al., 2017). It takes a future state that is $d$ time steps away from the current state as the hindsight / relabeled goal for the current state. When the goal is reached, a reward of 0 is given, otherwise a -1 reward is given for each time step.

In this case, we know it took $d$ steps to reach the hindsight goal, so the discounted future return is:

$$f(s_0) = \sum_{i=0,..,d-1} -1\gamma^i$$
$$= -1(1 - \gamma^d)/(1 - \gamma) \tag{5}$$

This calculation can be done on the fly as hindsight relabeling happens, requiring no extra space and very little computation.

We call this variant "lb-GD", short for lower bounding with goal distance based return.

Additionally, we can also apply lb-DR and lb-GD together, with discounted episodic return (lb-DR) on the original experience and goal distance based return (lb-GD) on the hindsight experience, giving the "lb-DR+GD" variant, which was used in Fujita et al. (2020).

### 3.2 In general (including non-episodic tasks)

If the task is continuing, without an episode end[3], discounted return needs to be accumulated all the way to infinity. When rewards are always non-negative, one can still use the accumulated discounted reward of the future n-steps to lower bound the value. But accumulated n-step discounted reward is no longer a lower bound when rewards can be negative, in which case, the more general lower bounding with bootstrapped value can be used: given a trajectory of training experience $\tau := <s_0, ..., s_n>$:

$$G_n^v(\tau) := r_1 + \gamma r_2 + ... + \gamma^{n-1} r_n + \gamma^n v(s_n) \tag{6}$$

Assuming the rewards and the state $s_n$ can be repeated with the same action sequence, $G_n^v(\tau)$ lower bounds the maximum achievable value $\bar{G}^v(s_0)$ (Equation 3).

Two variations are possible: Given a trajectory of length $n$,

1. compute $v(s_i)$ for all $i \in [1, n]$ and take the maximum of all $G_i^v(\tau)$ to obtain a tighter lower bound. We call this variant "lb-b-$n$step":

$$f(s_0) = \max_{i \in [1,n]} G_i^v(\tau) \tag{7}$$

2. only evaluate $v$ on the last ($n$th) step and use the $n$th-step bootstrapped return as the lower bound, which involves less compute but results in a looser bound. (When $n$ is large enough, this becomes the lb-DR variant.) We call this variant "lb-b-$n$step-only".

$$f(s_0) = G_n^v(\tau) \tag{8}$$

---

[2]Note that the behavior policy can be stochastic, as long as the policy class contains the optimal policy, value learning will converge to the optimal.

[3]Chapter 3.3 of Sutton & Barto (2018) has more details on episodic vs continuing tasks.

## 4   Integration into RL algorithms

### 4.1   Background

The value target lower bounds can be readily plugged into RL algorithms that regresses value to a target, e.g. DQN, DDPG or SAC.

In these algorithms, the action value $q(s, a)$ is learned through a squared loss with the target value $y$. In one step TD return, for a batch **B** of experience $\{s, a \rightarrow r, s'\}$, the loss is:

$$\mathcal{L}_q := \sum_{(s,a,r,s') \in \mathbf{B}} |q(s, a) - y|^2 \tag{9}$$

In one step TD return, $y$ is the one step TD return $\hat{q}(s, a, r, s')$:

$$\hat{q}(s, a, r, s') := r(s, a) + \gamma q'(s', \mu'(s')) \tag{10}$$

Here, $q'$ and $\mu'$ are the bootstrap value and policy functions, typically following the value and policy functions in a delayed schedule during training. (They are also called "target value" and "target policy", and are very different from the "value target" $y$ in this paper.)

### 4.2   Value target lower bounding

With lower bounding, we replace the value target $y$ with the lower bounded target:

$$y \leftarrow \max(f, \hat{q}(s, a, r, s')) = \max(f, r + \gamma q'(s', \mu'(s'))) \tag{11}$$

This way of lower bounding the value target is the same as was done by Fujita et al. (2020) (confirmed via personal communication), but is subtly and importantly different from lower bounding the $q$ value directly (Oh et al., 2018; Tang, 2020): $q(s, a) \leftarrow \max(f, q(s, a))$, which stays overestimated if $q(s, a)$ initially overestimates.

To the best of our knowledge, value target lower bounding with bootstrapped values is a novel contribution of this work.

### 4.3   An Illustrative Example

Figure 1 includes a fairly general example showing how value target lower bounding would improve value learning. Suppose we enhance an off policy algorithm such as DDPG with value target lower bounding (lb-DR), when there is no training experience hitting the target state, no meaningful training happens for the baseline or lb-DR. However, when there is one trajectory hitting the target state, all states along the trajectory will soon be propagated with meaningful return, and nearby states will also enjoy faster learning. As the state space becomes larger and the time horizon longer, a successful trajectory will speed up learning quite a bit.

## 5   Experiments

The goal is to demonstrate the sample efficiency of lower bounding the value target over baseline such as DDPG, TD3, SAC and HER. Because the lower bounded value target can now look potentially many steps into the future, we suspect it to be best suited for long horizon, sparse reward tasks. Hence, we choose to experiment on a sampled subset of Atari games, the goal conditioned FetchEnv tasks and the harder goal conditioned Pioneer Push and Reach tasks. See details of the experiment setup in Appendix 1.2.

### 5.1   Baselines

Baselines include DDPG (Lillicrap et al., 2016), TD3 (Fujimoto et al., 2018), SAC (Haarnoja et al., 2018) and HER (Andrychowicz et al., 2017; Plappert et al., 2018). Implementations are based on

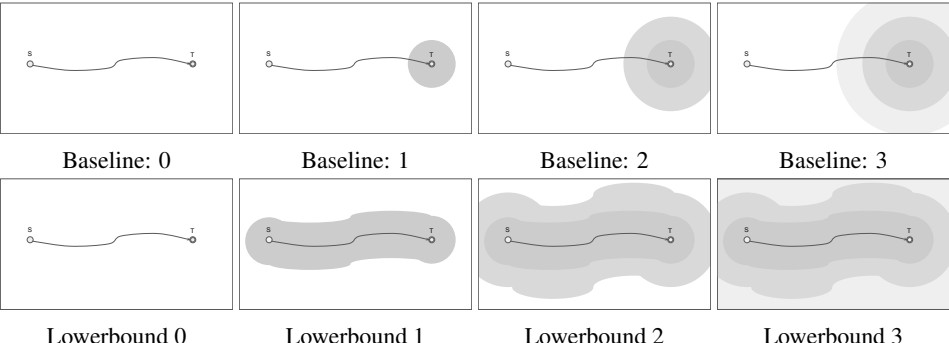

Figure 1: Illustration of value target lower bounding speeding up value learning as training progresses from stages 0 to 3. The task is to navigate in the state space from start state S to end state T, with sparse reward 1 at T and 0 elsewhere. The curve from S to T denotes a training experience that reaches the target. The shaded areas denote roughly states whose value has been significantly improved during training up to that stage.

open sourced repositories, and baseline performance is verified against published results under similar settings. The Appendix 1.6 and 1.5 include results on more baselines such as DDQN (van Hasselt et al., 2015), td-labmda (Sutton & Barto, 2018) and Retrace (Munos et al., 2016).

## 5.2  Hyperparameters

Value target lower bounding is applied on top of these baselines without any additional hyperparameter (Section 4). The only hyperparameters come from the baselines. These hyperparameters follow published work as much as possible. When baseline hyperparameters need to be tuned for an environment, e.g. Atari games or Pioneer tasks, we search for the best performance in total episode reward averaged across all tasks for that environment on one set of random seeds, then the optimal hyperparameters are fixed and evaluated on a separate set of random seeds never seen during development. Value target lower bounding simply uses the the parameter values optimal for the baselines. Details are in Appendix 1.3.

## 5.3  Results

We report results on both episodic and continuing/non-episodic tasks. We report evaluation performance averaged across several runs of the algorithms (five for the less stable Atari games and three for the others). Each run uses a random seed never seen during development. Due to space constraints, the main paper only reports performance aggregated across all tasks for each environment. During each run, we take one task and one random seed, run baseline and treatment algorithms, and record whether treatment agent evaluates strictly above the baseline agent as training progresses. We average across all the runs of the same environment, and plot the fraction of times where treatment is above baseline and the standard deviation of that fraction in Figure 2. Appendix 1.4 contains per task evaluation curves.

Overall, value target lower bounding is a simple, effective, efficient, carefree (no hyperparameter) and theoretically justified approach. Although the example lower bounds are limited to deterministic environments, the theory is generally applicable to stochastic environments. A similar prior work to compare would be Hindsight Experience Replay (HER) (Andrychowicz et al., 2017), which is simple, effective, efficient, and also limited to deterministic environments (Blier & Ollivier, 2021). However, unlike our work, HER relies on the task being goal conditioned with full knowledge of the reward function, has one hyperparameter to tune (the proportion of hindsight experience), and is not justified in theory for stochastic environments. Our work shows further significant gains on top of HER on hard continuous control tasks.

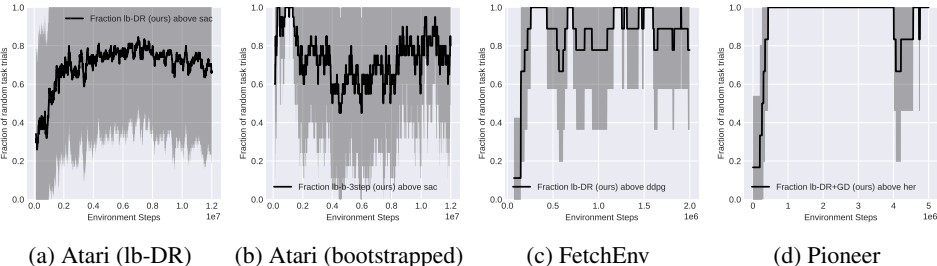

(a) Atari (lb-DR)  (b) Atari (bootstrapped)  (c) FetchEnv  (d) Pioneer

Figure 2: Aggregated evaluation performance: The fraction of times where treatment performs strictly above baseline, plotted along the number of time steps used for training. The solid curve is the sample mean of the fraction across all runs, and the shaded area is +/- one standard deviation. We use, for Atari (lb-DR), 85 runs – 17 games each with 5 seeds, for Atari (bootstrapped), 20 runs – 4 games 5 seeds, for FetchEnv, 9 runs – 3 tasks 3 seeds, and for Pioneer, 6 runs – 2 tasks 3 seeds.

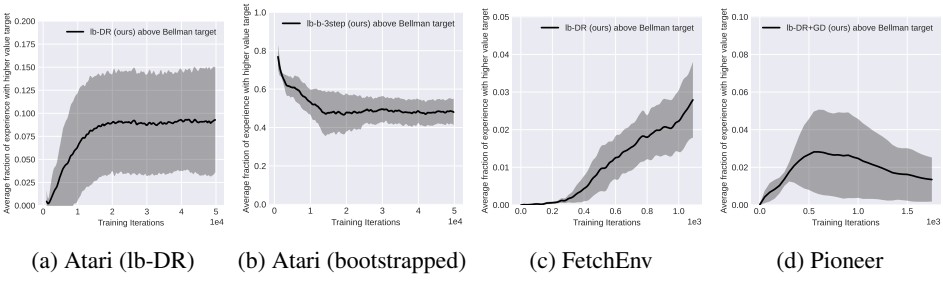

(a) Atari (lb-DR)  (b) Atari (bootstrapped)  (c) FetchEnv  (d) Pioneer

Figure 3: Average fraction of training experience where lower bounding improves Bellman value target, plotted along the number of iterations of training. The solid curve is the average of the fraction across all runs – number of tasks times number of seeds, and the shaded area is +/- one standard deviation. Other setups are the same as Figure 2.

### 5.3.1 Lower bounding vs baselines SAC/DDPG/HER

Figure 2 compares the lower bounding treatment with SAC, DDPG or HER baseline on 17 sampled Atari games, the FetchEnv tasks and the Pioneer tasks. For all the environments, value target lower bounding is not only more sample efficient, but also enjoys a higher converged performance. After training starts, it quickly gains ground and outperforms the baseline for 70% to 100% of the runs. It keeps that advantage even at the end of the training, outperforming baseline in converged performance. These plots show how frequently treatment is above baseline, but is insensitive to the magnitude of change.

Appendix 1.4 shows the magnitude of change with total episode return plotted for each task, often with large gains in sample efficiency and sometimes much higher converged performance. Among all the 22 tasks, only one task (Atari Breakout) shows lb-DR underperforming the baseline.

Investigations show that the loss on Atari Breakout is likely due to the mismatch between training objective and evaluation metric. During training, raw rewards are clipped to [-1, 1] and step discounted at $\gamma = 0.99$ to compute value, while in evaluation, total reward is the unclipped and undiscounted cumulative sum of episode rewards. The discount and clipping together severely penalizes large rewards earned later in the episode, which is what's happening for Breakout, because hitting a top layer block produces a reward of 7 while hitting a bottom layer block produces 1. When we use non-clipped rewards or a higher $\gamma$ in training, the lower bounding method performs much better in total reward. Note, this train-test discrepancy as well as an additional training bias (Thomas, 2014) is likely present in all of the prior works using policy gradient methods on Atari games.

### 5.3.2 Value target improvement

The lb-DR method is mostly effective, but is it really due to improvements to the value targets? Figure 3 looks at the fraction of training experience where lower bounding actually improves the

Bellman value target over the course of training. Overall, improved value target roughly coincides with performance gains. Appendix 1.4 shows the plots per each task.

The Appendix also has more results, comparing with baselines such as n-step methods, DDQN and optimality tightening, and more analyses such as ablations and robustness to hyperparameter choices.

## 6   Related Work

Prior works (Fujita et al., 2020; Hoppe & Toussaint, 2020; He et al., 2017; Oh et al., 2018; Tang, 2020) employed several different ways of computing future returns and using that as a lower bound to improve value learning. It is quite easy to introduce biases and inefficiencies into the process and end up with a suboptimal or inefficient algorithm. Our work is the first to propose the general form of value target lower bounding (possibly with bootstrapping), to show its convergence to the optimal value in the tabular case, and to demonstrate its effectiveness in illustrative examples and extensive experiments on a wide range of tasks.

Fujita et al. (2020)'s method is similar to a special case (the lb-DR+GD variant) of the general method. They used it as a part of a large system and showed that it improved sample efficiency for a robotic grasping task. Hoppe & Toussaint (2020) also bounded the value target. But instead of using empirical return, they used a simplified MDP with a subset of actions. Although without theoretical proof and only experimented on a limited set of robotic manipulation tasks, both works show that value target lower bounding increased sample efficiency. This work, in addition to the theory and the more general method, shows that lower bounding improves both sample efficiency and converged performance in a wide range of tasks.

He et al. (2017) used empirical return with bootstrap to improve value learning. They formulated value learning as a constrained optimization problem with the empirical bootstrapped value being the lower (and upper) constraints of the value function. In their experiments, the Lagrangian multiplier was fixed, which would likely lead to suboptimal solutions. Our lb-b-$n$step method also uses bootstrapped value. But we lower bound the value target directly, which is simpler, more efficient, and likely more optimal. Our work points out that for episodic tasks, even more efficient and effective methods like lb-DR exist. Appendix 1.5 offers more discussion and results related to this.

Our work is subtly but importantly different from the prior works on lower bound Q learning or Self Imitation Learning (SIL) (Oh et al., 2018; Tang, 2020). SIL uses empirical return $R$ to lower bound the value function itself (instead of the *value target*). This is achieved by adding an off policy value loss during on-policy (AC or PPO) training ($L_{value}^{sil} = \frac{1}{2}|v(s) - \max(v(s), R)|^2$). When the value function overestimates, the SIL value loss becomes zero, and keeps overestimating. Mixing the SIL loss with the loss from the baseline algorithms probably helped to correct the overestimation, but no theoretical guarantee was given. In evaluation, SIL was often compared to on-policy Actor Critic or PPO baselines, so it was not clear how much of the gain was due to lower bounding and how much due to off-policy value learning. In this work, we bound the Bellman value target (Equation 11), so overestimates are automatically corrected via Bellman updates, and convergence is guaranteed in the tabular case. We also use off-policy algorithms as baselines for a cleaner comparison.

N-step return methods such as td-lambda (Sutton & Barto, 2018) and Retrace (Munos et al., 2016) also look a few steps ahead, but to obtain more accurate value of the behavior policy. Traditionally, this requires careful off-policy correction, and the value can still be far from the optimal value due to the often suboptimal behavior. This work shows that value target lower bounding efficiently and effectively looks ahead much further without the need for off-policy correction, due to aiming at the optimal value. Appendix 1.5 has more detailed observations and discussions.

Planning methods can look into the future to achieve higher value targets and better control. Examples include Monte Carlo Tree Search (MCTS) (Schrittwieser et al., 2019; Ye et al., 2021) and Model Predictive Control (MPC) or receding horizon planning with raw actions (Chua et al., 2018; Hafner et al., 2019; Zhang et al., 2022), options (Silver & Ciosek, 2012), or subgoals (Nasiriany et al., 2019; Nair & Finn, 2020; Chane-Sane et al., 2021). Planning methods use either a dynamics model together with the learned value or just the learned value (in the case of goal conditioned tasks) (Nasiriany et al., 2019) to improve policy or value estimates. Planning typically happens during roll out (Nasiriany et al., 2019), but can also be used to improve the value target, as in Reanalyze of MuZero (Schrittwieser et al., 2019; Ye et al., 2021). During value improvement, if planning takes

the maximum over a set of possible future values (e.g. from different trajectories as in the case of MPC), and if this set includes the one step Bellman value target, then the planner is essentially using alternative trajectories and their values to lower bound the Bellman value target. In this sense, the theory developed here can potentially justify and improve Reanalyze. In general, planning is orthogonal to value target lower bounding, and typically requires additional components and a lot more compute than the basic TD learning does. Therefore, we leave it to future work to explore the synergy between the two.

Interestingly, it is common practice to lower and upper bound the returns to the possible region, e.g. Andrychowicz et al. (2017) bounds value between $[-\frac{1}{1-\gamma}, 0]$. Similar to lower bounding with episodic return (Section 3.1), such strict bounds of the actual value can be thought of as admissible heuristics (bounds) used during search of the optimal solution (Russell & Norvig, 2020). What's new in this work is that lower bounding with bootstrapped values (which can overestimate the value) still converges to the optimal value.

Kumar et al. (2020) (DisCor) also recognized that bootstrapped value targets can be inaccurate. This bias impacts learning adversely under function approximation. DisCor uses distribution correction to sample experience with accurate bootstrap targets more frequently, while value target lower bounding aims to directly reduce the bias.

While in theory using empirical return to lower bound the value target is only correct for deterministic environments, in practice, it seems as long as the environment is not heavily impacted by random fluctuations, they still perform well. In fact, with function approximation, the agents cannot distinguish between two slightly different states, making the problem partially observable (Sutton & Barto, 2018) and appear slightly random. Prior methods such as SIL (Oh et al., 2018), Optimality Tightening (He et al., 2017), and even Hindsight relabeling (Andrychowicz et al., 2017) and MuZero (Schrittwieser et al., 2019) require the environment to be deterministic. Despite this theoretical limitation, the lower bounding methods and the prior methods can still be very useful, outperforming baselines often by large margins and when deploying to the real world (Fujita et al., 2020).

# 7 Conclusions

We propose a general form of lower bounding the value target using possibly bootstrapped return. In theory, value target lower bounding converges to the same optimal solution as the original Bellman operator. In practice, several ways of finding value lower bounds are examined.

For episodic tasks, discounted episodic return is an efficient and effective method involving very little extra computation. Precomputing the episodic return and storing it into the replay buffer allows efficient lower bound computation. It achieves much higher sample efficiency and converged performance than one-step baselines such as SAC, DDPG or TD3 in most tasks, and is competitive among n-step baselines. Simple goal distance based return uses even less compute and achieves large gains in certain long horizon tasks over Hindsight relabeling (HER).

For non-episodic tasks or in general, lower bounding with n-step bootstrapped return outperforms one-step baselines and is a strong competitor to the n-step methods such as (truncated) td-lambda and Retrace.

## 7.1 Future Work

There are probably better ways of finding value lower bounds that improve training even more. One direction may be to use planning (e.g. Monte Carlo Tree Search, the Cross Entropy Method or using subgoals) to achieve tighter lower bounds given a model of the task.

Estimating value lower bound for stochastic tasks may be possible, e.g. by learning a reward function and a dynamics model and using imagined rollouts to obtain bootstrapped returns without overestimation.

Other ways of bounding the value target, e.g. upper bounding (He et al., 2017), may be worth investigating as well, e.g. to reduce overestimation in regions of poor reward.

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
