# 1 Appendix

## 1.1 Proofs

Before going on to Theorem 2.3, we first prove the easier Corollary 2.4.

We want to show that given a lower bound of the optimal value, $\forall s, f(s) \leq \mathcal{B}^\infty(v)(s)$, under the new operator $\mathcal{B}_f$, the value function converges to the same optimal value function given by the Bellman operator $\mathcal{B}$.

*Proof.* Let $v^*$ be the fixed point and optimal value of the original Bellman operator: $v^* := \mathcal{B}^\infty(v)$, $v$ be any value function, and $s$ any state.

First, for the simple case of $\forall s$, where $f(s) \leq \mathcal{B}(v)(s)$, the new operator backs off to the Bellman operator, and follows the convergence of the Bellman operator:

$$
\begin{aligned}
&|\mathcal{B}_f(v)(s) - v^*(s)| \\
=&|\max(\mathcal{B}(v)(s), f(s)) - v^*(s)| \\
=&|\mathcal{B}(v)(s) - v^*(s)| \\
=&|\mathcal{B}(v)(s) - \mathcal{B}(v^*)(s)| \\
\leq&\gamma||v - v^*||_\infty
\end{aligned}
$$

Second, $\forall s$, where $f(s) > \mathcal{B}(v)(s)$,

$$
\begin{aligned}
&|\mathcal{B}_f(v)(s) - v^*(s)| \\
=&|\max(\mathcal{B}(v)(s), f(s)) - v^*(s)| \\
=&|f(s) - v^*(s)| \\
=&v^*(s) - f(s) \quad \text{(because } f \text{ lower bounds } v^* : v^*(s) \geq f(s)) \\
<&v^*(s) - \mathcal{B}(v)(s) \quad \text{(because } f(s) > B(v)(s)) \\
=&|\mathcal{B}(v)(s) - v^*(s)| \\
\leq&|\mathcal{B}(v)(s) - v^*(s)| \\
=&|\mathcal{B}(v)(s) - \mathcal{B}(v^*)(s)| \\
\leq&\gamma||v - v^*||_\infty
\end{aligned}
$$

Therefore, the distance to the optimal value shrinks by $\gamma$ with every application of $\mathcal{B}_f$:

$||\mathcal{B}_f(v) - v^*||_\infty = \max_s |\mathcal{B}_f(v)(s) - v^*(s)| \leq \gamma||v - v^*||_\infty$.

According to the definition of convergence to $v^*$, we need to find an $N$, such that $\forall \epsilon > 0, \forall v \neq v^*$, $\forall n > N, ||\mathcal{B}_f^n(v) - v^*||_\infty < \epsilon$.

We can easily calculate that any $N \geq \log_\gamma \frac{\epsilon}{||v-v^*||_\infty}$ (note, $\gamma < 1$) satisfies the condition, which concludes the proof that any value function $v$ will converge to $v^*$ under the lower bounded Bellman operator $\mathcal{B}_f$. $\square$

Note, from the proof above, we can see that $\mathcal{B}_f$ converges faster than $\gamma$ (the speed of Bellman contraction), when the lower bound is strictly above the Bellman value target, i.e. $f(s) > \mathcal{B}(v)(s)$.

Now for Theorem 2.3: given the maximum achievable value $\bar{G}^v(s)$ and given that $f(s) \leq \bar{G}^v(s)$, we want to show convergence to the optimal value.

*Proof.* First, $\forall s$, where $f(s) < \mathcal{B}(v)(s)$, the value target backs off to the original Bellman target, and the distance to the optimal value shrinks at rate $\gamma$.

Second, $\forall s$, where $f(s) < v^*(s)$, it follows from Corollary 2.4 that the distance to the optimal value shrinks at rate $\gamma$.

Last, we only need to prove for any $s$, where $f(s) \geq v^*(s)$ and $f(s) \geq \mathcal{B}(v)(s)$, the distance to the optimal value still shrinks:

$$|\mathcal{B}_f(v)(s) - v^*(s)|$$
$$=|\max(\mathcal{B}(v)(s), f(s)) - v^*(s)|$$
$$=|f(s) - v^*(s)|$$
$$=f(s) - v^*(s)$$
$$\leq \bar{G}^v(s) - v^*(s)$$

According to the definition of $\bar{G}^v$ in Equation 3 of the main text:

$$= \max_{\pi \in \Pi, n \in [1, +\infty)} G_n^{\pi, v}(s) - v^*(s)$$

Now suppose $\pi'$ and $n(s)$ achieves the maximum bootstrapped value $\bar{G}^v(s)$:

$$= G_{n(s)}^{\pi', v}(s) - v^*(s)$$

According to the definition of n-step bootstrapped value $G_n^{\pi, v}(s)$ in Equation 2 of the main text:

$$= \mathbb{E}^{\pi'}\{r_1 + \gamma r_2 + ... + \gamma^{n(s)-1} r_{n(s)} + \gamma^{n(s)} v(s_{n(s)})\} - v^*(s)$$

(The expectation above is over all possible $n(s)$-step trajectories of the given policy $\pi'$ and MDP.)

Suppose $\pi^*$ is the optimal policy, which achieves maximum value $v^*$ for any number of steps $n$ and state $s$:

$$= \mathbb{E}^{\pi'}\{r_1 + \gamma r_2 + ... + \gamma^{n(s)-1} r_{n(s)} + \gamma^{n(s)} v(s_{n(s)})\} - \mathbb{E}^{\pi^*}\{r_1 + \gamma r_2 + ... + \gamma^{n(s)-1} r_{n(s)} + \gamma^{n(s)} v^*(s_{n(s)})\}$$
$$\leq \mathbb{E}^{\pi'}\{r_1 + \gamma r_2 + ... + \gamma^{n(s)-1} r_{n(s)} + \gamma^{n(s)} v(s_{n(s)})\} - \mathbb{E}^{\pi'}\{r_1 + \gamma r_2 + ... + \gamma^{n(s)-1} r_{n(s)} + \gamma^{n(s)} v^*(s_{n(s)})\}$$

(Inequality holds because $\pi^*$ maximizes the expected n-step value bootstrapped with the optimal value $v^*$.

Thus, the expected value of a different policy, e.g. $\pi'$, bootstrapped with $v^*$ will be smaller or equal.)

$$= \gamma^{n(s)} \mathbb{E}^{\pi'}\{v(s_{n(s)}) - v^*(s_{n(s)})\}$$
$$\leq \gamma^{n(s)} \mathbb{E}^{\pi'}|v(s_{n(s)}) - v^*(s_{n(s)})|$$
$$= \gamma^{n(s)} \sum_{s_{n(s)}} \{p^{\pi'}(s_{n(s)}|s) \times |v(s_{n(s)}) - v^*(s_{n(s)})|\}$$
$$\leq \gamma^{n(s)} \max_s |v(s) - v^*(s)|$$
$$\leq \gamma^{\min_s n(s)} \max_s |v(s) - v^*(s)|$$
$$= \gamma^{\min_s n(s)} ||v - v^*||_\infty$$

This means in the case of overestimated bootstrap values, the new operator promises to shrink at a rate of $\gamma^{\min_s n(s)}$, and overall, the new operator will at least shrink at a rate of $\gamma$. $\qquad\square$

Note, from the proof above, we can see that when the lower bound overestimates, i.e. $f(s) \geq v^*(s)$, $\mathcal{B}_f$ converges at a speed of $\gamma^{\min_s n(s)}$, which could be faster than $\gamma$, the speed of Bellman contraction.

These proofs work for stochastic MDPs, because we treat trajectory rewards and states as random variables conditioned on the MDP and the policy. The proofs work for action values as well, by simply replacing the value function above $v(s)$ with the action value $q(s, a)$, and the value lower bound $f(s)$ with the action value lower bound $f(s, a)$.

$\pi'$ in theory can be different for different state $s$, so that when unrolling from state $s_0$ for a few steps into $s_i$, it still follows $\pi'(s_0)$, instead of $\pi'(s_i)$. However, it's easy to prove (by contradiction) that there exists a single policy $\pi'$ which achieves the maximum achievable value (as long as ties are split deterministically).

## 1.2 Experiment Setups

We experiment on three sets of tasks with different input characteristics and control difficulty. Some of the tasks are not goal conditioned, so only lower bounding with empirical discounted return is available. Some of them are goal conditioned, so both empirical discounted return and hindsight relabeling with discounted goal return as lower bound are available.

### 1.2.1 Atari games

We experiment on the classical Atari games with image input to test using discounted episodic return to lower bound value target. We picked the popular games Breakout, Seaquest, Space Invaders (these three were used for hyperparameter tuning by Mnih et al. (2013)), Atlantis, Frostbite and Q*bert (these three were highlighted in (He et al., 2017)), and randomly sampled another 11 games from the total 56. As with prior work (Mnih et al., 2013; Oh et al., 2018), we evaluate on the deterministic versions of the games (NoFrameskip-v4) with actions repeated for a fixed four frames and each game started with up to 30 random noop steps before handing to the agent. Each episode of the games is capped at 10,000 time steps, with the last time step having discount 0.99 when the time limit is reached, i.e. resetting the game without ending the episode. For a regular game end after losing all lives, the episode ends, i.e. last step discount is 0. Because the games are episodic, both lb-DR and lb-b-$n$step methods can be applied.

### 1.2.2 Fetch Push, Slide and PickAndPlace

The FetchEnv tasks (Plappert et al., 2018) are goal conditioned tasks with a robotic arm moving objects on a table. Robot states and object position serve as input. The agent outputs continuous actions taking the form of relative positions to move to. A PID controller translates the relative position actions into the exact torque applied at each joint. Rewards are sparse and goal-conditioned, with -1 for non-goal states and 0 for goal states.

By default the FetchEnv tasks are non-episodic. They reset every 50 steps, but all steps including the step right before task reset have the same positive discount (Andrychowicz et al., 2017). As explained in Section 3.1, to use episodic discounted return as lower bound, we can make them episodic by adding a gym wrapper around the environment to end an episode after its goal is achieved, and reset the task. When a goal is not reached within 50 steps, we just reset the task without ending the episode, as is done in the original FetchEnv, and such experience is not used in value target lower bounding.[1] This also changes the nature of the tasks, so the agent does not have to stay at the goal state indefinitely, but instead only needs to reach the goal position as fast as possible. This makes the episodic FetchEnv tasks slightly easier to train than the original tasks, because the agent only needs to reach the goal state quickly, instead of having to reach and stay at the goal position indefinitely. (There are ways to avoid changing the desired behavior by e.g. including agent's speed into the goal state or requiring the agent to stay at the goal position for several time steps before ending the episode. This seems orthogonal to the main idea here, and is not included in this work.)

We experiment on both the original/non-episodic FetchEnv tasks (with lb-b-$n$step methods) and the episodic FetchEnv tasks (with lb-DR and lb-GD methods).

Compared with the Atari games, the inputs are simpler, no longer image based, but the control task is continuous, under realistic physical simulation and harder.

### 1.2.3 Pioneer Push and Reach tasks

This is a set of challenging goal reaching and object pushing tasks for the physically simulated car Pioneer 2dx. The car is 0.4 meter long. Objects and goal positions are randomly initialized between 0.5 meter to 1 meter of each other inside a 10 meter by 10 meter flat space. Inputs are the car and object states and the goal positions, and actions are the forces applied on the two driving wheels.

For the Pioneer Push task, the car has to push a block to within 0.5 meter of the 2 dimensional goal position indicated by a small red dot on the ground. For the Pioneer Push and Reach task, the car has to first push the object to the goal location (red dot) and then drive to a separate goal position (red ball in the air); the goal is achieved when the concatenation of the two goal locations (for Push and for Reach) is within 0.5 of the concatenated achieved positions (of the block and the car) in $L_2$-distance.

These tasks are episodic with sparse goal reward, and we only test the lb-GD and lb-DR+GD methods on them with HER as baseline. (TD3 without HER takes too long to train.)

---

[1] Fujita et al. (2020) chose to end episodes when either a maximum of T time steps is reached or the goal is reached, and provided the agent with the number of timesteps since episode start as input to the agent, so that the agent is aware of the approaching episode end.

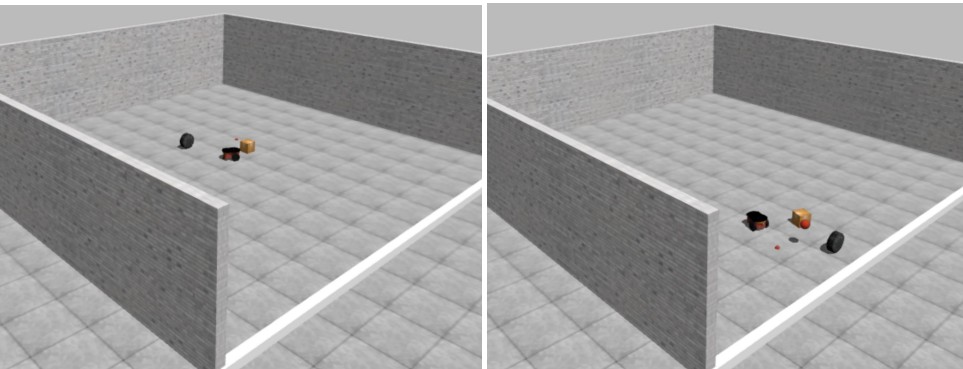

Figure 1: The Pioneer Push task and the Push and Reach task.

These tasks take longer time to accomplish, and also take longer time to train than the FetchEnv tasks. Some of the reasons are the force based wheel control instead of the higher level position control, and the arena space being much larger than just a tabletop.

## 1.3 Hyperparameters

Table 1 lists the hyperparameters of the baseline algorithms. For FetchEnv, they follow published work (Plappert et al., 2018). For Atari and Pioneer tasks, they are tuned using one set of random seeds and after keeping the hyperparameters fixed, trained with a different set of random seeds and evaluated. Value target lower bounding has no parameter, and we did not re-tune any parameters of the baseline RL algorithms for value target lower bounding. When comparing lb-b-$n$step methods with other n-step methods, we simply use the same $n$ as the other baselines.

For the Atari games, the original DQN setup with only one training environment takes too long to train so we decided to tune SAC as baseline and found it to outperform published Actor-Critic results (Oh et al., 2018) and our tuned DDQN (results in Appendix 1.6).

For Pioneer Push and PushReach tasks, TD3 is used, (we simply equip DDPG with two critics for clipped double Q learning (Fujimoto et al., 2018)), which works better than DDPG with one critic.

Table 1: Hyperparameters for all the tasks

| Hyperparameters\Tasks | Atari (SAC) | FetchEnv (DDPG) | Pioneer (TD3) |
|---|---|---|---|
| Parallel environments | 30 | 38 | 30 |
| Unrolls per train iteration | 8 | 50 | 100 |
| Updates per train iteration | 4 | 40 | 40 |
| Mini-batch size | 500 | 5,000 | 5,000 |
| Training updates per target network update | 20 | 40 | 40 |
| Target update weight | 0.95 | 0.95 | 0.95 |
| Discount per time step | 0.99 | 0.98 | 0.99 |
| Initial collect steps | 100,000 | 10,000 | 10,000 |
| Total training time steps | 12 million (x4 environment frames) | 2 million | Push: 5 million, PushReach: 14 mil |
| Max steps before task reset | 10,000 | 50 | Push: 100, PushReach: 200 |
| Replay buffer size | 1 million | 2 million | 6 million |
| Adam optimizer learn rate[a] | $5e^{-4}$ | $1e^{-3}$ | $1e^{-3}$ |
| Network structure | conv((32, 8, 4), (64, 4, 2), (64, 3, 1)) + fc(512)[b] | fc(256, 256, 256) | fc(256, 256, 256) |
| Number of critics | 2 | 1 | 2 |
| $\epsilon$-greedy for evaluation | 0.05 | 0.3 | 0.3 |
| Evaluation interval in train iters | 1000 | 40 | 200 |
| Evaluation episodes | 100 | 200 | 100 |
| Life loss as terminal[c] | Yes | n/a | n/a |
| Action repeat | 4 | n/a | n/a |
| Image scaling | [-1, 1] | n/a | n/a |
| Frame stacking | 4 | n/a | n/a |
| Reward clipping | [-1, 1] | n/a | n/a |
| SAC target entropy | calculated[d] | n/a | n/a |
| Hindsight percentage | n/a | 80% | 50% |
| Observation normalization[e] | No | Yes | No |
| TD-lambda: $\lambda$ | 0.95 | 0.95 | n/a |
| $n$-step bootstrap: $n$ | 3 | 2 | n/a |

[a] Adam $\hat{\epsilon} = 1e^{-7}$ for all tasks.

[b] Netowrk structure for Atari follows DDQN (van Hasselt et al., 2015).

[c] Life loss in Atari games is treated as a terminal state in training, following EfficientZero (Ye et al., 2021).

[d] The SAC target entropy is set to the entropy of uniformly distributing 0.1 probability mass across all but one actions.

[e] Observations are normalized to have zero mean and unit variance based on the statistics of the training observations, and we found the normalization to be critical in reproducing HER results on FetchEnv.

## 1.4 Results

Applying lower bounding (e.g. lb-DR) on different baseline algorithms e.g. DDPG or SAC results in different treatment methods. Since we always compare treatment with its corresponding baseline, throughout the paper, we simply call the treatment lb-DR etc. without mentioning the baseline algorithm.

### 1.4.1 lb-DR (episodic return) vs baseline SAC/DDPG

Figure 2 compares lower bounding with discounted return (lb-DR) against SAC or DDPG baseline on 17 sampled Atari games and the episodic FetchEnv tasks.

For 16 out of the 17 Atari games, lower bounding with episodic discounted return (lb-DR) performs at least as well as the baseline, often much better. On more than half of the Atari games and on the Fetch PickAndPlace task, there are large gains in both sample efficiency and final performance. On FetchPush and a few of the Atari games (Alien, Bank Heist and Fishing Derby), there is about 70% sample efficiency gain with similar converged performance. Among all the 20 tasks, only 1 task (Atari Breakout) shows lb-DR underperforming the baseline.

1.4.1.1    Value learning plots

This section presents plots of learned value and how often value is improved by the proposed methods, in order to show the effect of lower bounding on value improvement.

Figure 3 shows the fraction of training experience where lb-DR value target is greater than the Bellman target from SAC/DDPG on the 17 Atari games and the episodic FetchEnv tasks (without hindsight). They correlate well with actual performance (Figure 2) and with how value is learning (Figure 4).

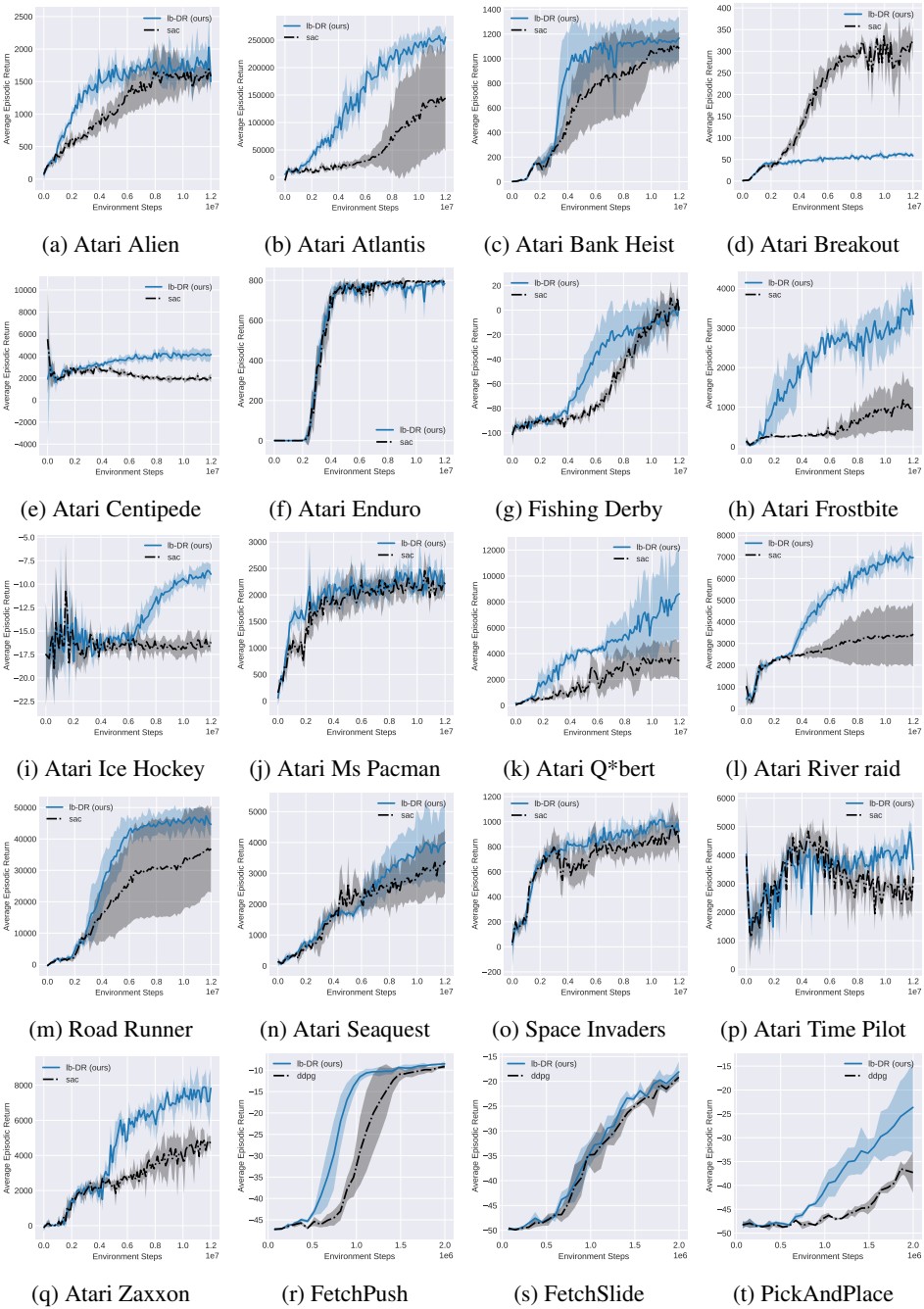

Figure 2: Evaluated average return of value target lower bounding with discounted return (lb-DR) vs SAC or DDPG on Atari games and episodic FetchEnv tasks. Solid curves are the mean across five (for Atari) or three (others) seeds, and shaded areas are +/- one standard deviation.

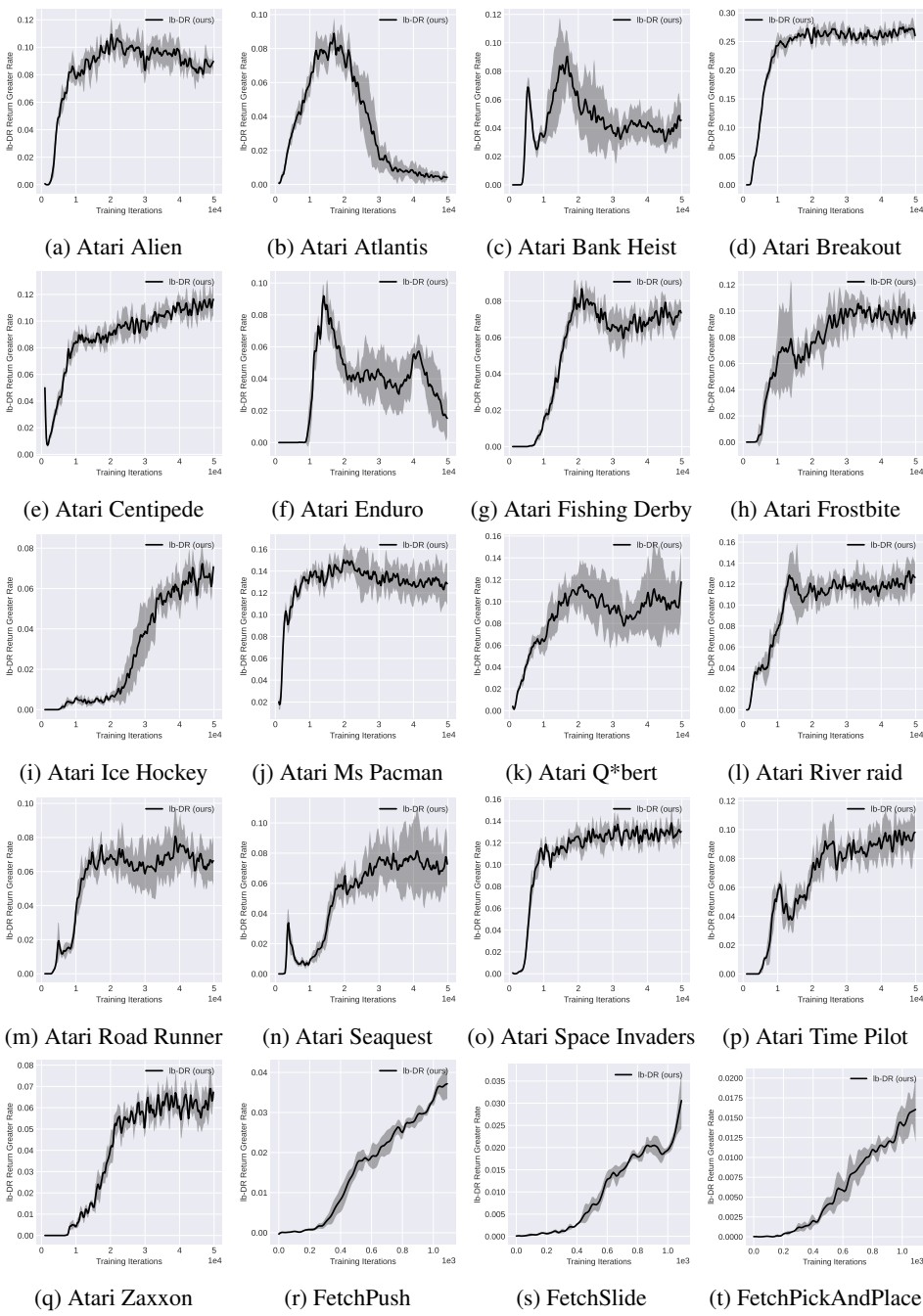

Figure 3: Fraction of training experience where lb-DR value target is greater than the Bellman target, on Atari games and episodic FetchEnv tasks, plotted against the number of training iterations. Solid curves are the mean across five (for Atari) or three (others) seeds, and shaded areas are +/- one standard deviation.

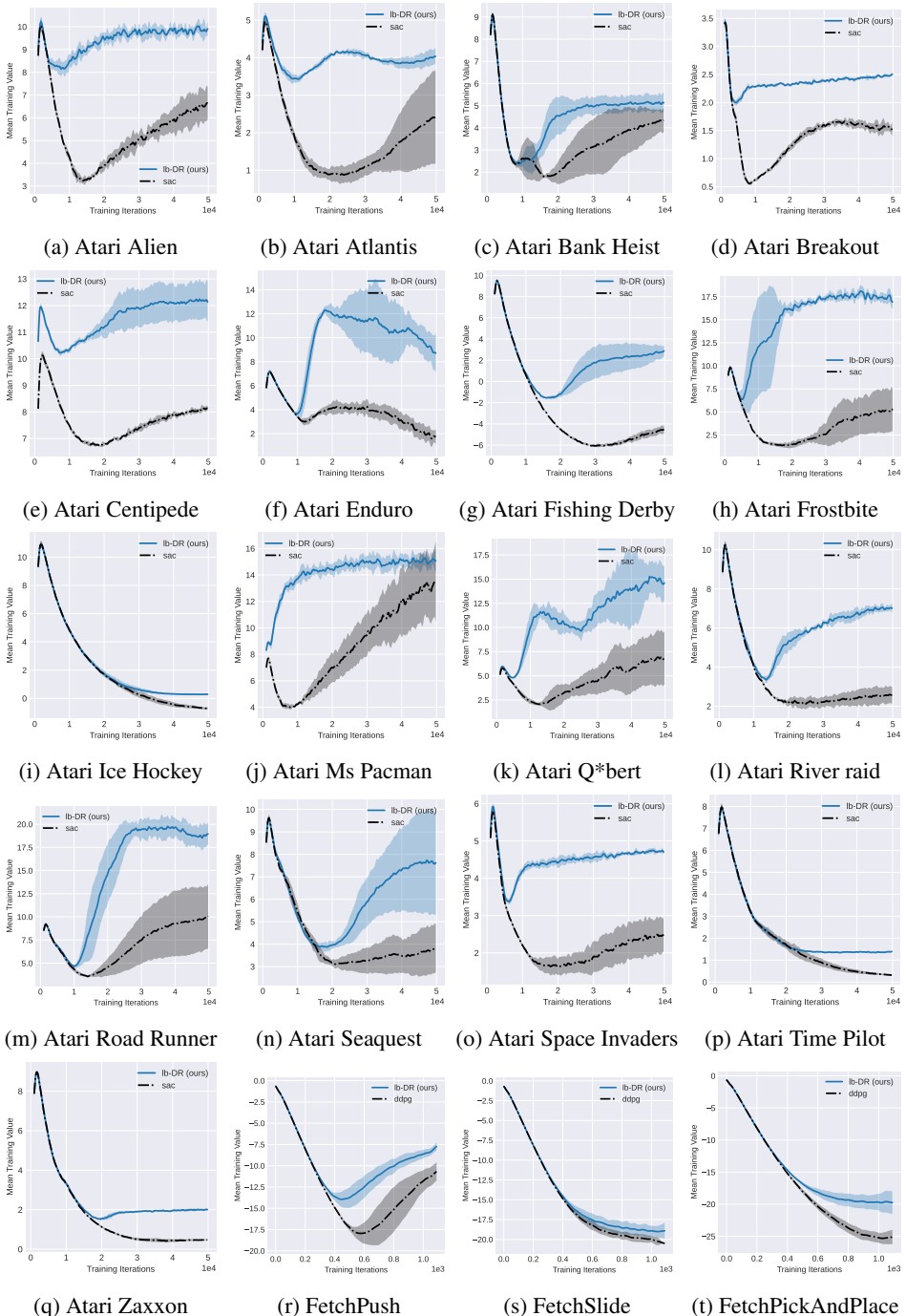

Figure 4: Learned values of lb-DR and SAC (for Atari games) and DDPG (for FetchEnv tasks), evaluated on the training experience and plotted against the number of training iterations. Solid curves are the mean across five (for Atari) or three (others) seeds, and shaded areas are +/- one standard deviation.

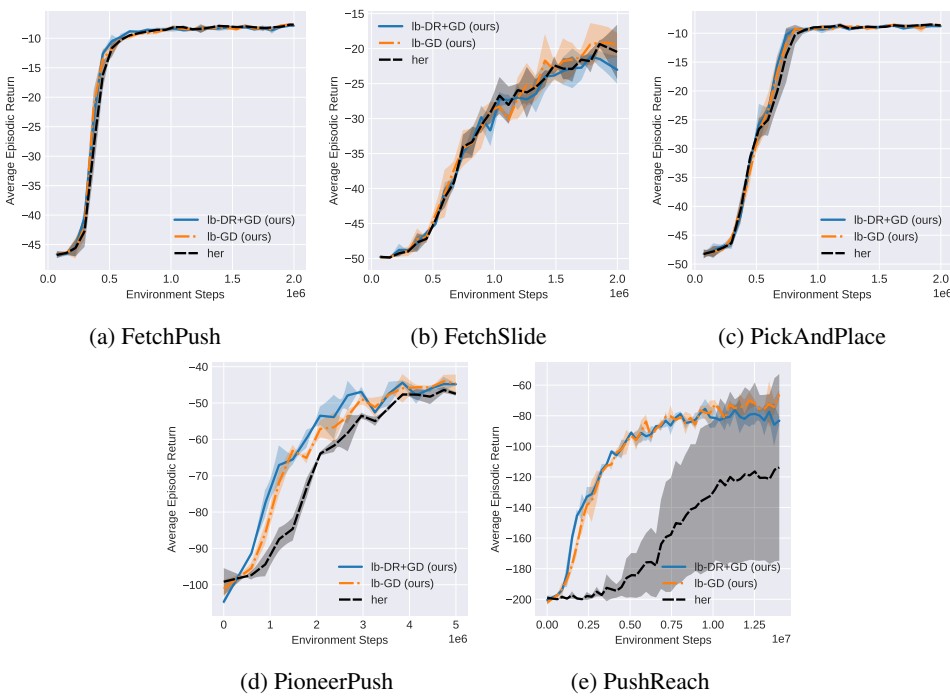

(a) FetchPush  (b) FetchSlide  (c) PickAndPlace

(d) PioneerPush  (e) PushReach

Figure 5: Value target lower bounding with goal distance return (lb-GD) and lb-DR+GD vs HER on episodic FetchEnv and Pioneer tasks. Solid curves are the mean across three seeds, and shaded areas are +/- one standard deviation.

129 **1.4.2 lb-GD (goal distance return) and lb-DR+GD vs HER**

130 Figure 5 compares lower bounding with goal distance return (lb-GD) and lower bounding with both
131 goal distance and discounted return combined (lb-DR+GD) against the much stronger HER baseline,
132 on the goal conditioned episodic FetchEnv and Pioneer tasks.

133 On the easier FetchEnv tasks, lower bounding is similar as HER, but on the more challenging Pioneer
134 Push and Reach tasks, lower bounding is able to achieve over 70% more sample efficiency. It seems
135 the harder the task, the wider the margin of gain.

136 1.4.2.1  Value learning plots

137 This section presents plots of learned value and how often value is improved by the proposed methods,
138 in order to show the effect of lower bounding on value improvement.

139 Figure 6 shows the fraction of training experience where the lb-GD is higher than the Bellman value
140 target from HER, in the goal conditioned (episodic FetchEnv and Pioneer) tasks, and the learned value.
141 It seems, for FetchEnv tasks, where lb-GD only performs slightly better than HER, the fraction of
142 experience with improved value target is quite small (less than 1%). Hindsight relabeling is probably
143 already producing fairly high value targets. For Pioneer Push and Reach tasks, lb-GD performs much
144 better in average return, and the fraction of experience with higher value target is also much larger
145 (peaking around 2-8%).

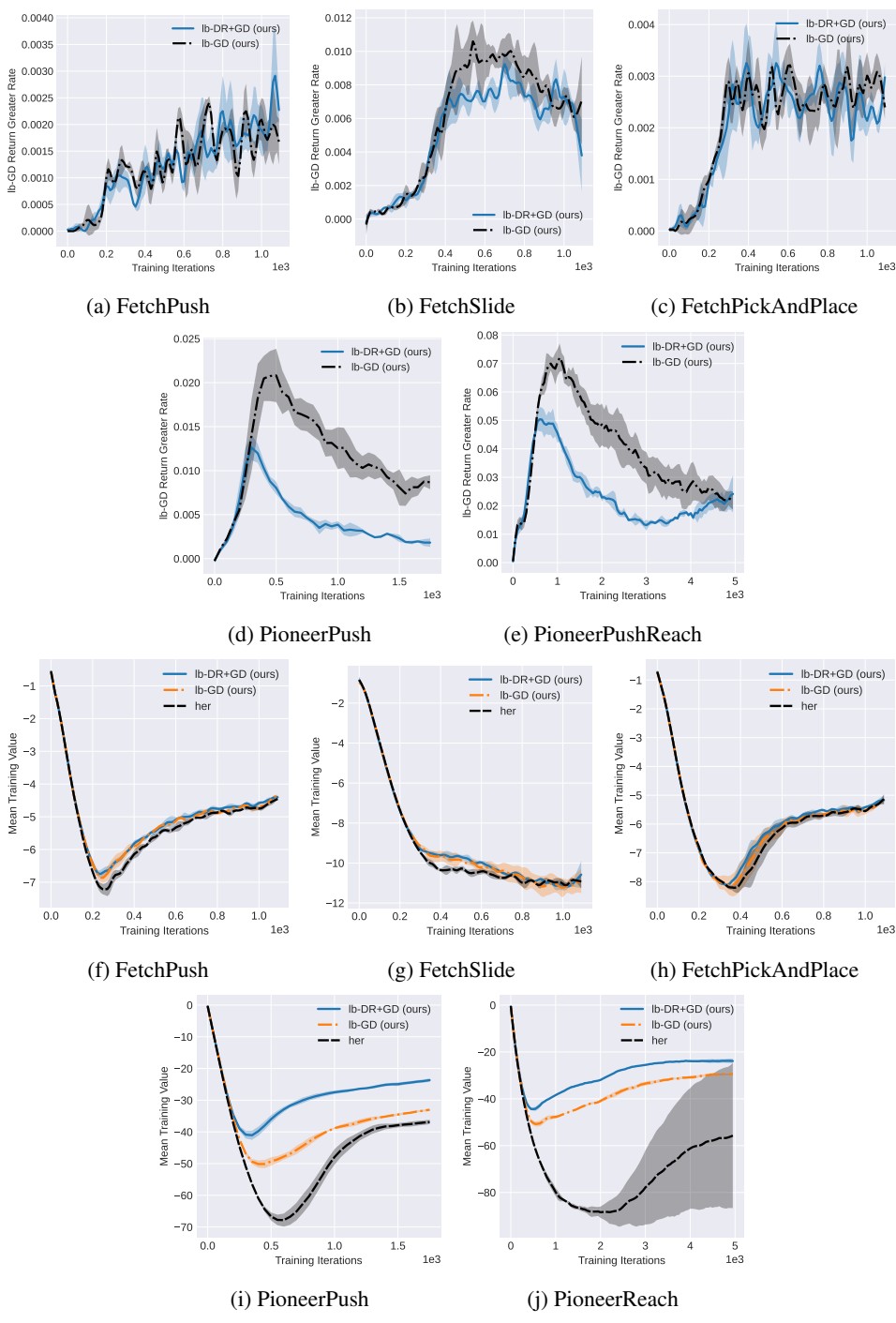

Figure 6: Fraction of training experience where lb-GD or lb-DR+GD value target is greater than the Bellman target (a-e) and learned values (f-j), on episodic FetchEnv and Pioneer tasks, plotted against the number of training iterations. Solid curves are the mean across three seeds, and shaded areas are +/- one standard deviation.

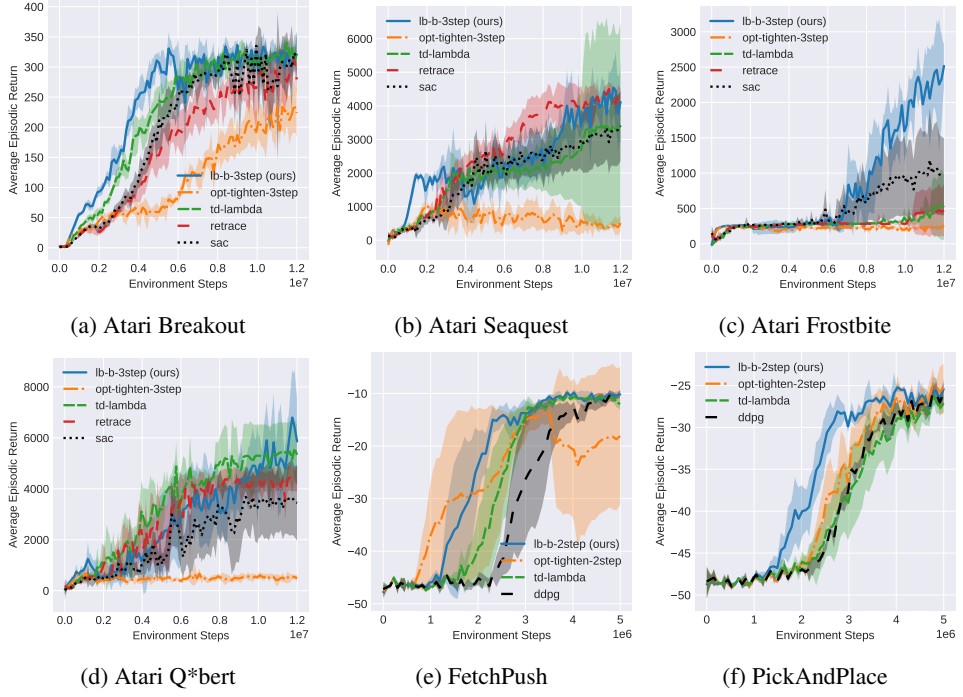

Figure 7: Evaluated average return of value target lower bounding with n-step bootstrapped return (lb-b-$n$step) vs SAC or DDPG, td-lambda, Retrace and optimality-tightening on Atari games and the original/non-episodic FetchEnv tasks. $n = 3$ for Atari and 2 for FetchEnv tasks. Solid curves are the mean across five (for Atari) or three (others) seeds; shaded areas are +/- one standard deviation.

### 1.4.3 n-step bootstrapped methods

Figure 7 shows performance of lb-b-$n$step methods on a subset of the Atari games (episodic) and the original (non-episodic) FetchEnv tasks. Besides SAC/DDPG, baselines also include n-step methods such as td-lambda, Retrace (Munos et al., 2016) and optimality tightening (He et al., 2017). lb-b-$n$step methods are at least as good as the best baseline method, and clearly outperforms the baselines in two of the six tasks.

#### 1.4.3.1 Value learning plots

Figure 8 shows the fraction of experience where lb-b-$n$step lower bounds are above one step Bellman value targets. There seems to be a correlation between improving the value target over more experience and actually improving the policy, at least for FetchEnv tasks. The correlation does not seem as clear as that of lb-DR. With bootstrapping, the fractions are generally higher than those of lb-DR in Figure 3, potentially due to overestimated bootstrap values.

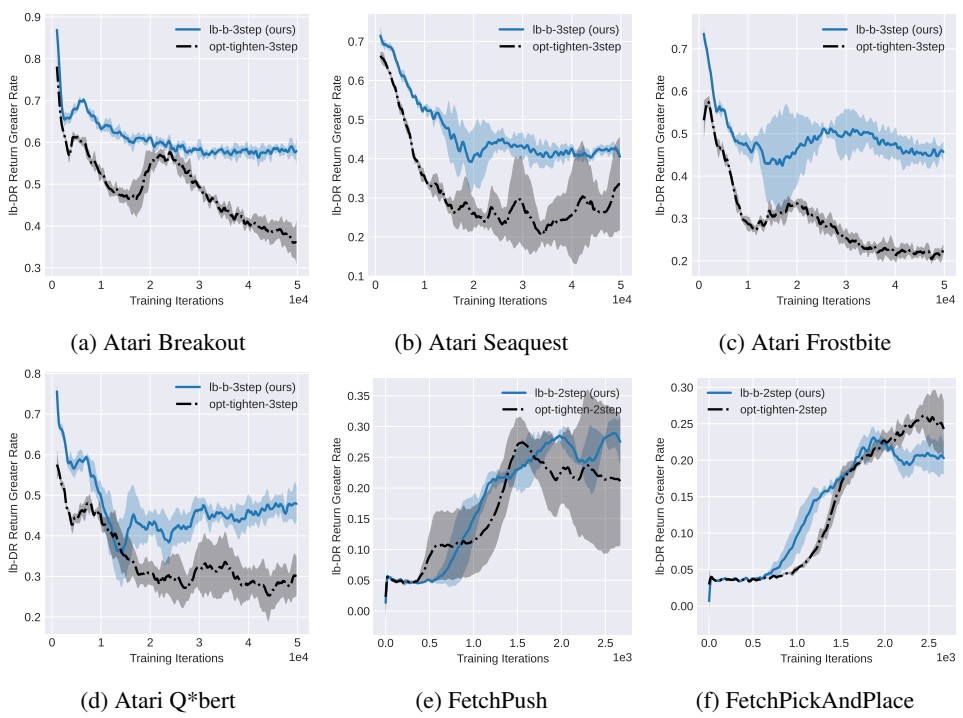

Figure 8: Fraction of training experience where lb-b-$n$step value target or optimality-tightening lower bound is greater than the value target of baselines SAC (for Atari games) and DDPG (for FetchEnv tasks), plotted against the number of training iterations. Solid curves are the mean across five (for Atari) or three (others) seeds, and shaded areas are +/- one standard deviation.

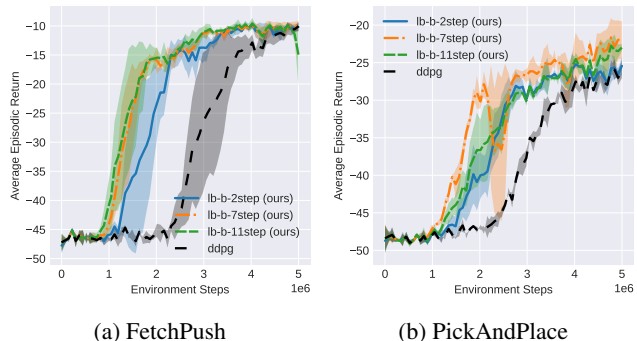

(a) FetchPush          (b) PickAndPlace

Figure 9: Evaluated average return of value target lower bounding with n-step bootstrapped return (lb-b-$n$step) with different $n$ on the original/non-episodic FetchEnv tasks. Solid curves are the mean across three seeds, and shaded areas are +/- one standard deviation.

#### 1.4.4    n-step method ablations

The following three sections include ablations which show the lower bounding methods to be robust to variations in the hyperparameters.

1.4.4.1    lb-b-$n$step with different $n$ (number of steps)

Figure 9 shows the effect of how the number of steps $n$ in $n$-step bootstrapped return impacts lower bounding performance. The lb-b-$n$step method is not very sensitive to the value $n$, typically, the higher the value $n$ the better the performance, while n-step methods like td-lambda or Retrace would degrade a lot as $n$ increases above 3 or 4. We also observe lower value overestimation as $n$ increases (Figure 13).

1.4.4.2    lb-b-$n$step vs lb-b-$n$step-only (only n-th step)

Figure 10 shows the effect of taking a maximum of all 2- to n-step bootstrapped returns versus only using the n-step bootstrapped return. It seems using the maximum bootstrapped return of all 2- to n-steps, hence a tighter lower bound, works better than only using the n-step return.

1.4.4.3    lb-b-$n$step (bootstrap) or lb-DR (episodic return)

In continuing tasks (with negative rewards), we have to use the bootstrapped lb-b-$n$step method. But for episodic tasks, should we use bootstrapped return or episodic return as value target lower bound? In theory, lb-b-$n$step-only becomes lb-DR when $n$ is large enough. In practice, in terms of effectiveness, we can compare lb-b-$n$step with lb-DR on the Atari games (Figure 7 and 2 respectively). lb-b-$n$step is better than lb-DR on Atari Breakout. On Seaquest, the two are similar. On the other two games: Frostbite and Q*bert, episodic return is better. It seems lb-DR is better on tasks where rewards are more sparse and longer term planning is needed. In terms of efficiency, as $n$ becomes larger, the memory and compute efficient lb-DR method will become more attractive. Overall, both methods show a clear advantage over the baselines.

1.4.4.4    Value learning plots

Figure 11 shows the fraction of experience where lb-b-$n$step lower bounds are above one step Bellman value targets.

Figures 12 and 13 show the learned value of the lb-b-$n$step and lb-b-$n$step-only methods. It's convenient to look at the FetchEnv tasks which should always have non-positive value. From Figure 13(a,b), it seems as $n$ increases, value decreases, maybe due to more accurate estimates of value. From Figure 13(c,d), it seems the tighter lower bounds of lb-b-$n$step method do lead to slightly more overestimation in value.

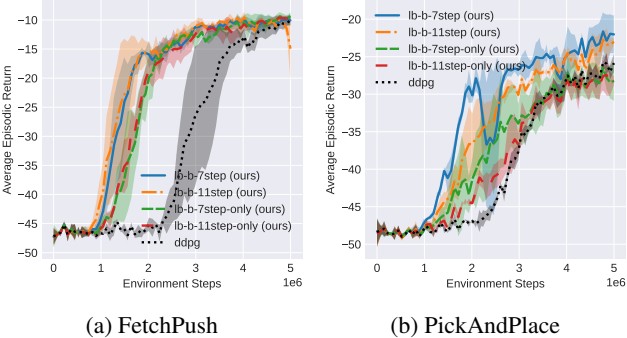

(a) FetchPush

(b) PickAndPlace

Figure 10: Evaluated average return of value target lower bounding with all n-step bootstrap (lb-b-$n$step) and nth-step only (lb-b-$n$step-only) on the original/non-episodic FetchEnv tasks. Solid curves are the mean across three seeds, and shaded areas are +/- one standard deviation.

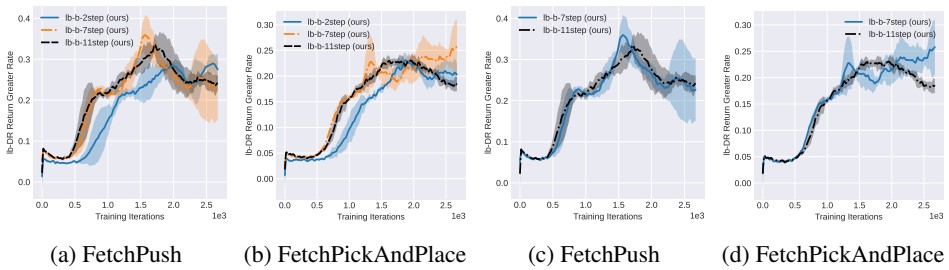

(a) FetchPush    (b) FetchPickAndPlace    (c) FetchPush    (d) FetchPickAndPlace

Figure 11: Fraction of training experience where lb-b-$n$step or lb-b-$n$step-only improves over the baseline Bellman value target, evaluated on the training experience and plotted against the number of training iterations. Solid curves are the mean across three seeds, and shaded areas are +/- one standard deviation.

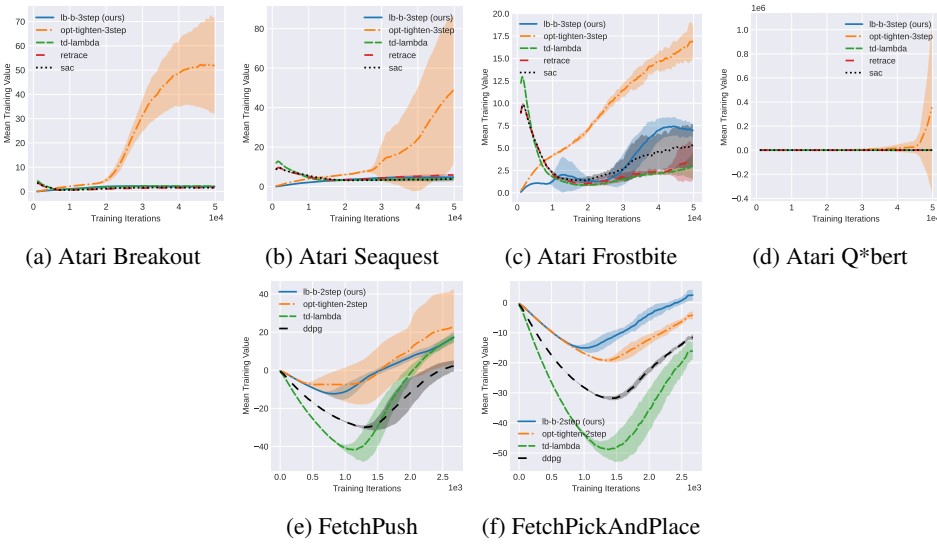

(a) Atari Breakout    (b) Atari Seaquest    (c) Atari Frostbite    (d) Atari Q*bert

(e) FetchPush    (f) FetchPickAndPlace

Figure 12: Learned values of lb-b-$n$step and SAC (for Atari games), DDPG (for FetchEnv tasks), optimality-tightening, td-lambda and Retrace, evaluated on the training experience and plotted against the number of training iterations. Solid curves are the mean across five (for Atari) or three (others) seeds, and shaded areas are +/- one standard deviation.

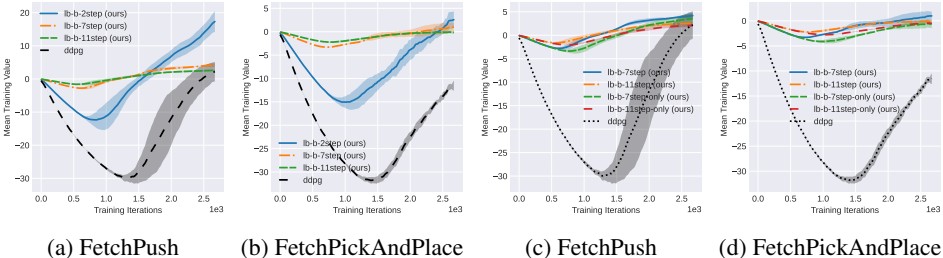

(a) FetchPush     (b) FetchPickAndPlace     (c) FetchPush     (d) FetchPickAndPlace

Figure 13: Learned values of lb-b-$n$step, lb-b-$n$step-only, DDPG (for FetchEnv tasks), optimality-tightening, td-lambda and Retrace, evaluated on the training experience and plotted against the number of training iterations. Solid curves are the mean across three seeds, and shaded areas are +/- one standard deviation.

## 1.5  N-step return based methods

### 1.5.1  N-step return methods

lb-b-$n$step methods and n-step return methods are similar in their data and computation requirements, and we already compare their performance in Figure 7.

Here, we additionally compare lb-DR directly with n-step methods in Figure 14.

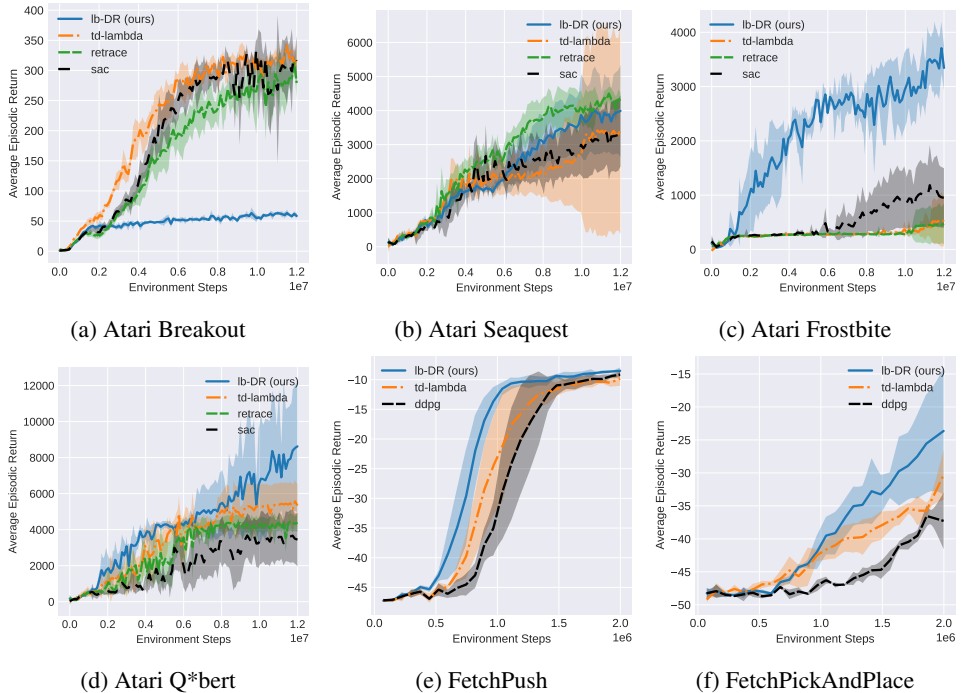

Figure 14: Evaluated average return of value target lower bounding with discounted return (lb-DR) vs SAC or DDPG, td-lambda and Retrace on four Atari games and the episodic FetchEnv tasks. Solid curves are the mean across five (for Atari) or three (others) seeds, and shaded areas are +/- one standard deviation.

During experimentation, we found that n-step methods are typically harder to tune and more expensive to compute.

1) Tuning $n$: A small $n$ for n-step methods works similarly as the baseline one-step method, and a larger n hurts performance. This is likely due to the off-policy bias in n-step return causing the n-step estimate to be potentially worse than the one-step estimate. Introducing importance sampling weights (Retrace) would help reduce the bias, but at the same time significantly downweight the off-policy high return experiences, making an ineffective use of such experiences.

None of these issues exist in value target lower bounding: (a) It does not incur any off-policy bias, and (b) as long as an experience renders high reward, being off-policy does not affect its ability to improve the value target.

2) Tuning involves other hyperparameters like the td-lambda parameter, replay buffer size, prioritized replay (to potentially expire old experiences and sample recent ones more frequently), target network update parameters (to reduce potential overestimation), and parameters for importance sampling. But still, after all the tuning, it only slightly outperforms one-step DDPG on FetchEnv or SAC on Atari games, and is often below the lower bounding methods. For td-lambda and Retrace, the best performance comes from 2-step (for FetchEnv) or 3-step (for Atari) td with $\lambda = 0.95$, all other parameters the same as the baseline DDPG or SAC. Retrace underperforming the baseline in Breakout and Frostbite is similarly observed in the original paper (Munos et al., 2016).

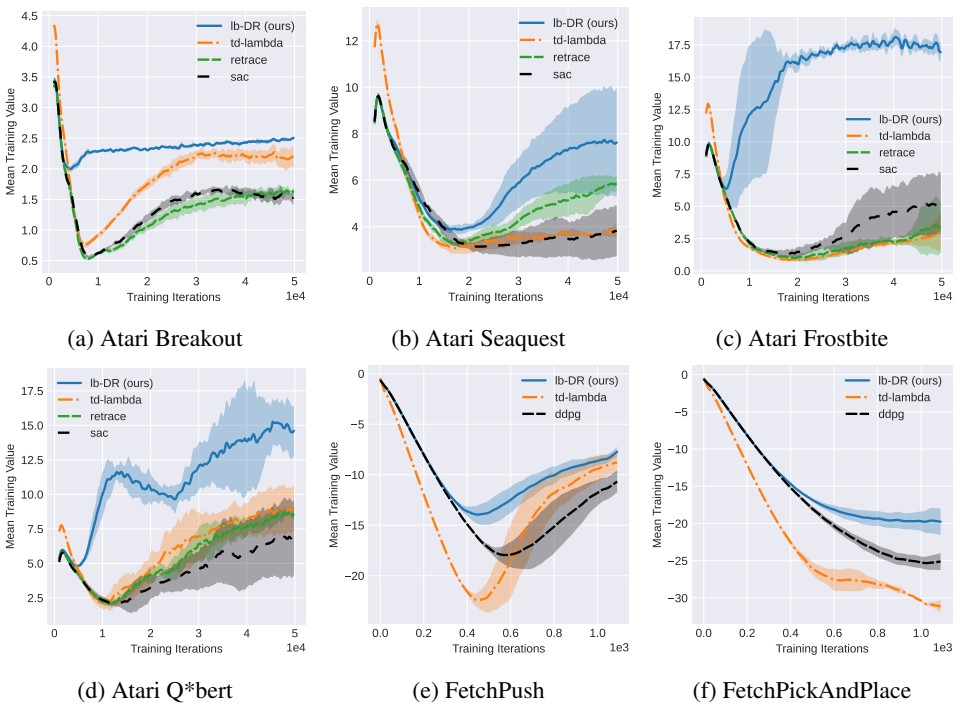

Figure 15: Learned values of lb-DR and SAC (for Atari games), DDPG (for FetchEnv tasks), td-lambda and Retrace, evaluated on the training experience and plotted against the number of training iterations. Solid curves are the mean across five (for Atari) or three (others) seeds, and shaded areas are +/- one standard deviation.

On the other hand, value target lower bounding with episodic return requires no hyperparameter tuning, (lb-b-$n$step with bootstrapping is not very sensitive to the choice of $n$), and learns faster on most tasks and converges higher on many tasks.

3) Computing n-step td-lambda return requires more computation due to evaluating value networks on all n-steps of the experience. It limits how large $n$ can be due to GPU memory limits, and slows down training time significantly with a large $n$.

On the other hand, value target lower bounding precomputes and stores episodic discounted return in the replay buffer, and incurs very little additional computation.

Overall, n-step methods are much more expensive and difficult to use, and the much simpler and effective lower bounding methods maintain an advantage in both effectiveness and efficiency. We show the performance comparison in Figure 14 with learned values in Figure 15.

### 1.5.2 Optimality tightening with n-step returns

He et al. (2017) use bootstrapped n-step return to lower and upper bound the value during training. They frame the problem as a constrained optimization problem where the distance between the value and the Bellman value target is minimized subject to the constraints that the value function must be within the lower (and upper) bounds. This is similar to our lb-b-$n$step method, except instead of applying a constraint on the value, we use the bootstrapped return to directly lower bound and improve the value target, which is likely more optimal and more efficient. In their experiments, the Lagrangian multiplier was fixed, which would likely lead to suboptimal solutions, and no theoretical guarantee was given. For episodic tasks, even more efficient and effective methods like lb-DR exist.

Some detailed differences:

1) The prior work bounds the value function itself (similar to lower bound q learning (Oh et al., 2018; Tang, 2020)), instead of bounding the Bellman value target. This could cause suboptimal training because the Bellman target itself could be outside the bounds, causing contradictory training targets

and losses. Imagine the current value for a state is 1, its Bellman value target may be a low 0, and the lower bound may be a high 2, then it's unclear which way the value function should go. It will depend largely on the mixing weight between the two losses $\lambda$ and whether initial values overestimate, which can be hard to tune in practice. In their experiments, a fixed Lagrangian multiplier $\lambda$ was used, which makes the method likely non-optimal.

2) In order to compute the bootstrapped values, the value network needs to be evaluated on all n future time steps, severely increasing GPU memory consumption and compute. Because of this increase in compute, in experiments, it could only look at a limited (4) timesteps into the future, while lb-DR and lb-b-$n$step-only can look ahead much further with very little extra computation and storage.

We implemented the method (He et al., 2017) (only using the lower bounds, no upper bounds) and integrated into our baselines. We ran on FetchPush and FetchPickAndPlace with hyperparameters number of time steps $n = 2$, and the penalty coefficient $\lambda = 4$ following the original paper. Results in Figure 7 show optimality tightening to be either not as stable or not as optimal as our lb-b-$n$step. We also ran it on Atari games, and found that optimality tightening overestimates value a lot, leading to much worse behavior than even the SAC baseline. The constrained optimization formulation might have an adverse interaction with RL training, and the upper bounds in the original optimality tightening work may be necessary to bring training back on track. As future work, it will be interesting to see whether value target upper bounding with bootstrapped values (as proposed in (He et al., 2017) as value upper bound constraints) can lead to optimal converged value in theory, and whether the upper bounds would help optimality tightening in experiments.

### 1.6 Value target lower bounding on DDQN

Because DQN is a more popular baseline for the Atari games, we've also applied value target lower bounding (lb-DR) on DDQN, and ran on a subset of the Atari games. Different from (van Hasselt et al., 2015), our implementation of DDQN finds the maximal action using the target critic network and evaluates the target value on the target network. It also uses two critic replicas like in (Fujimoto et al., 2018). Figure 16 shows the results. Compared to SAC (Figure 2), DDQN either lowers the baseline performance or the treatment (lb-DR) performance, and does not seem as strong as SAC.

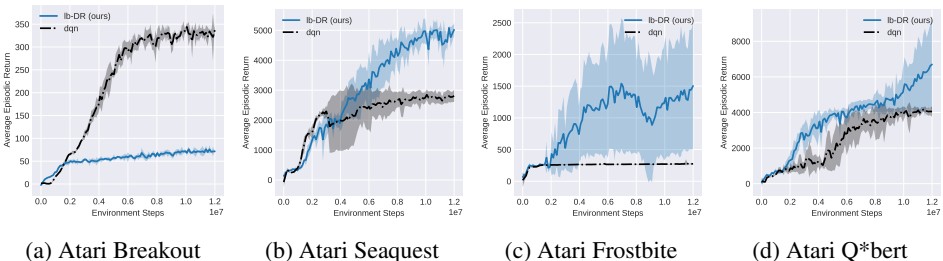

(a) Atari Breakout      (b) Atari Seaquest      (c) Atari Frostbite      (d) Atari Q*bert

Figure 16: Evaluated average return of value target lower bounding with discounted return (lb-DR) implemented on DDQN vs DDQN baseline on four of the Atari games. Solid curves are the mean across five seeds, and shaded areas are +/- one standard deviation.

### 1.7 A stochastic example

Using empirical return directly as value lower bound can lead to value overestimation under stochastic MDPs, as shown in the example below.

Assume state $S_0$ always goes to $S_1$, and $S_1$ gets reward $\pm 2$ 50% of the times when transitioning to the terminal state. Then $v(S_0) = v(S_1) = 0$. However, with value target lower bounding, for the lucky case with reward 2, the value target for $S_0$ is $\gamma \max(2, v(S_1)) = 2\gamma$, and for the unlucky case with reward -2, the value target for $S_0$ is $\gamma \max(-2, v(S_1)) = \gamma v(S_1) = 0$. On average, $v(S_0)$ will be overestimated to be $\gamma$.

It is worth noting that lower bounding the action value directly as done in SIL (Oh et al., 2018) will overestimate not just $v(S_0)$ but $v(S_1)$ as well, whereas lower bounding the value target will produce

the correct $v(S_1)$. This is because the same trajectory is used to both produce the Bellman value target ($\pm 2$ for $S_1$) and the lower bound ($\pm 2$ for $S_1$) which will be exactly the same for a given trajectory.

The more advanced MuZero algorithm (Schrittwieser et al., 2019) also has the limitation to deterministic environments. For this particular stochastic example, MuZero can still produce unbiased estimates of $v(S_0)$ due to averaging across two step rollouts from $S_0$.

Despite this theoretical limitation, value target lower bounding in practice still performs quite well over baselines when using function approximation, which does not perfectly represent the states and adds randomness to the tasks.

## 1.8 Potential improvement

Note that the goal distance based return (lb-GD) of Section 3.1.1 is a very simple way of arriving at a reasonable lower bound with near zero additional computation. The bound could be made tighter. Typically, an $L_2$ distance threshold is used to judge goal achievement, which will likely be satisfied a few time steps before exactly arriving at the hindsight goal. To compute such a tighter bound would require evaluating the reward function across the trajectories of experience using all possible hindsight goal states, and storing them in the replay buffer, i.e. episode length squared more computation and more storage space. It may be worth doing when episodes are short, or doing it only for a small number of time steps into the future when e.g. rewards are non-negative.

## 1.9 Societal Impact

This research belongs to basic reinforcement learning, accelerating the training of reinforcement learning algorithms. Helpful agents trained on benevolent tasks will learn faster, so will malicious agents trained to do damage. All the usual societal impact of reinforcement learning algorithms apply. The reader and user has to use caution and their own judgments when applying these automation algorithms to the real world: careful testing and scaling is needed, starting from virtual simulations, to testing in a lab environment, to small scale real world tests, and eventually to full scale deployment with careful monitoring in place.

## Reproducibility Statement

Our code change is based on a publicly available RL library, with strong baselines already implemented. Our relatively small code change is committed to a private github repository, which we plan to open source upon publication. Experiment parameters are configured and controlled by an automation script, with each experiment label corresponding to the set of configurations used for that experiment, so there is little room for manual error when running many experiments across different tasks, methods and hyperparameters. When running experiments, the snapshot of the code used to run each experiment is stored together with the results for verification.

Experiments are done in simulation with pseudo randomness. We've run our code on different machines with different GPU hardware using the same docker image, and the results are reproducible up to every float number using the same random seed. In a few cases, we've also run our code on different versions of hardware and software (CUDA and pytorch), and the results are similar, though not the same at the float number level.