# OpenReview forum: "Faster Reinforcement Learning with Value Target Lower Bounding"
_NeurIPS.cc/2022/Conference — NeurIPS 2022 Submitted_

### Official Review · Reviewer_5ybC · 2022-06-26

**Rating:** 3
**Confidence:** 4
**Soundness:** 2 fair
**Presentation:** 3 good
**Contribution:** 2 fair

**Summary:**

This paper proposes a form of target value lower bounding, which induces a new Bellman operator. The authors show the operator is a contraction, and propose several lower bounds. Extensive experiments are conducted (provided in the supplementary material) to show the performance of the proposed approach.

**Questions:**

- For the proposed lower bounds - are the theoretical gaps between the deterministic and non-deterministic settings?
- When is $\bar{G}$ more beneficial than $G_\infty$
- Why does $B_f$ improve convergence?
- How does $B_f$ affect the overestimation problem?

**Limitations:**

The authors discuss limitations of their work. Particularly, deterministic requirements.

**Strengths And Weaknesses:**

This paper is written clearly and the overall concept is interesting. There are several aspects of this paper that are problematic.

First, theoretically, the paper is not strong. Theorem 2.3 does not present any interesting new theory, and it is unclear why this result is important. The authors do not show or prove any benefit of using the bounded Bellman operator. When would this operator be useful for cases that don't bound $G_\infty$? It would seem that due to over-optimism of the Bellman operator, $G_0$ will usually maximize $\bar{G}$.

Second, the examples for lower bounds in the paper require deterministic environments. The lower bounds themselves feel somewhat trivial. It is unclear why these lower bounds are good.

Finally, the experiments do not show any significant improvement using the proposed approach. Also, the experiments themselves are shown only in the appendix, which should not be used to place major contributions. The authors chose to do this due to space constraints, which is not fair to other reviewed papers.

Strengths:

- Paper is clearly written
- Proposed Bellman operator may be having interesting characteristics to be researched
- Authors conducted extensive experiments

Weaknesses:

- Lack of theory, and unsure about novelty
- Setting is limited. Examples require deterministic environments. Unsure why $\bar{G}$ is useful.
- Experiments don't show a clear benefit of approach. Also, experiments are only provided in appendix which does not need to be evaluated for review.

---

> ### Author Response · Authors · 2022-07-30
> **Reply to Reviewer 5ybC**
>
> Thanks for the careful review and detailed comments.
>
> 1. Regarding the value of the theory: This paper aims to show that value target lower bounding is an effective and theoretically justified approach.  Theorem 2.3 justifies the use of n-step bootstrapped return as lower bound, and Corollary 2.4 justifies the use of episodic return as lower bound, that the lower bounded Bellman operator will result in optimal converged value.  This result is novel to the best of our knowledge, and without the theory, erroneous uses of lower bounding can lead to detrimental results in learning.  For example, prior works on directly lower bounding the value instead of the value target do not correct overestimations, and the converged values can be wrong.
>
>    $\bar{G}$ is defined as the maximum over all bootstrapped returns achievable by any policy, which makes Theorem 2.3 a stronger and more general result, in particular, it encompasses Corollary 2.4.
>
> 2. Regarding empirical benefit vs limitation to deterministic environments: This work aims to present a simple, effective, efficient, carefree (no hyperparameter) and theoretically justified approach, although the example lower bounds are limited to deterministic environments.  A similar prior work to compare is Hindsight Experience Replay (HER), which is simple, effective, efficient, and also limited to deterministic environments [1].  However, unlike our work, HER relies on the task being goal conditioned with full knowledge of the reward function, has one hyperparameter to tune, and is not justified in theory for stochastic environments (our theory is generally applicable to stochastic environments, only the example lower bounds have the limitation).  Our work shows further significant gains on top of HER on hard continuous control tasks.  (We included this discussion in the updated main paper.)
>
> 3. Regarding why the seemingly simple lower bounds are helpful: An illustrative example in Section 4.3 shows when learning can be faster.  Empirical results on many tasks (Section 5.3) show that lower bounding with the example lower bounds improves both converged performance and sample efficiency.
>
> 4. Regarding experimental gains: We’ve included summary plots of evaluation results in the updated main paper, showing gains in both sample efficiency and converged performance.  Thanks for bringing it up.
>
> 5. We are not exactly sure what Question 1 was about.  The proposed lower bound, e.g. empirical episodic return, is a valid lower bound of the value only when the task is deterministic.  In a stochastic setting, some episodes can yield high returns by chance, resulting in overestimations of value and no longer a lower bound.  Does this address your question?
>
> 6. Section 5.3.2.4 (before the current update, and Appendix 1.4.4.3 after update) explains when $\bar{G}$ could be a more beneficial or practical lower bound than $G_{\infty}$.
>
> 7. $B_f$ improves convergence when the lower bound is closer to the optimal value than the Bellman value target is.  See the illustrative example in Section 4.3 and the proofs in Appendix 1.1 when strict inequalities hold.  We included a discussion of this in the appendix of the new draft, right after the proofs.
>
> 8. $B_f$ results in the optimal value as long as conditions of Theorem 2.3 are satisfied, in particular, $f$ is a lower bound of the maximum achievable value, regardless of whether the current value function $v$ or the lower bound $f$ overestimates or not.
>
> [1] Léonard Blier and Yann Ollivier, Unbiased Methods for Multi-Goal Reinforcement Learning, CoRR 2021 https://arxiv.org/abs/2106.08863

---

### Official Review · Reviewer_3Knv · 2022-07-08

**Rating:** 3
**Confidence:** 4
**Soundness:** 2 fair
**Presentation:** 1 poor
**Contribution:** 2 fair

**Summary:**

This paper suggests that learning a lower bound of the value function can facilitate dynamic programming such enhance RL performance. The author(s) provided theoretical insights about the optimality of the value function learning with their lower bounding method. It was claimed the proposed methods showed effectiveness in a wide range of tasks.


**Questions:**

How to implement your method in deep RL? Because Algorithm 1 is for tabular case.


**Limitations:**

See Weaknesses.

**Strengths And Weaknesses:**

Strengths:

1. The proposed method seems simple but somehow effective.
2. Novel theoretical insights about lowerbounding the value function.
3. The author(s) carefully explained the novelty of the proposed method in Sec.6.


However, I could not evaluate the effectiveness of the proposed methods because all the empirical results are in the appendix. Meanwhile, as I have shortly scanned the appendix, the are also other things to notice, which I will details below.

I think the manuscript in its current form is not acceptable as a NeurIPS conference paper. Because the important methods (how to implement in deep RL) and experimental results are all in the appendix, as well as some other problems exist. If the authors(s) cannot include all the essential methods and results in a conference paper, I recommand to submit to a journal.

Weaknesses:

1. As I said, the paper should be self-contained with essential practical algorithms and experimental results in its main texts.
2. Estimating value function has always been an vital problem in deep RL. Although the paper discussed related work about lower bounding the value function, there are other studies pointed out that Q-learning like algorithms, e.g., SAC, tend to over-estimate the value targets , which is bad and should be regularized. For example, [1][2][3] used different methods to control the over-estimating bias of SAC and observed very large performance gain. The paper should discuss the relation to them and conduct comparison to at least one of them.


minor issue I found:

line 140, "action value" --> "state-action value"


[1] Kuznetsov, Arsenii, et al. "Controlling overestimation bias with truncated mixture of continuous distributional quantile critics." International Conference on Machine Learning. PMLR, 2020.

[2] Chen, Xinyue, et al. "Randomized Ensembled Double Q-Learning: Learning Fast Without a Model." International Conference on Learning Representations. 2021.

[3] Hiraoka, Takuya, et al. "Dropout Q-Functions for Doubly Efficient Reinforcement Learning." International Conference on Learning Representations. 2022.

---

> ### Author Response · Authors · 2022-07-30
> **Reply to Reviewer 3Knv**
>
> Thanks for the careful review and detailed comments.
>
> 1. The original draft explains implementation in the RL (non-tabular) setup in Section 4, which the reviewer may have missed?  We’ve also included summary plots of evaluation results in the updated main paper, showing gains in both sample efficiency and converged performance.  Thanks for bringing it up.
>
> 2. Regarding value overestimation of baseline RL algorithms, lower bounding with an overestimated function could further increase overestimation, and the methods [1][2][3] could be used to further improve performance.  However, we feel that reducing overestimation is orthogonal to the proposed lower bounding, and may be more similar to value target upper bounding.  Furthermore, for Atari games, SAC already takes the minimum value of two critic networks, and we did not observe any obvious overestimation in our experiments.  For FetchEnv tasks, double critic may be too conservative, and slows learning down (tried, but not reported in the paper).

---

> > ### Comment · Reviewer_3Knv · 2022-08-08
> > **Thanks for the reply**
> >
> > Thanks for updating the manuscript. I have updated my rating. However, I still lean toward rejection given the comments of other reviewers, and because of the lack of comparison. As the authors said, [1][2][3] may be more similar to value target upper bounding. It would be more convincing and if the authors can compare with them or similar works. Furthermore, I encourage the authors to combine their method with the technique of value target upper bounding to see whether upper and lower bounding work good together.

---

### Official Review · Reviewer_zNRT · 2022-07-10

**Rating:** 5
**Confidence:** 4
**Soundness:** 2 fair
**Presentation:** 3 good
**Contribution:** 3 good

**Summary:**

The paper proposes value target lower bounding, where a function lower bounds the maximum achievable value is used as the target value.  Such a function admits the same convergence with the original Bellman operator but converges faster. The proposed value target lower bounding is claimed to be general and can be combined with any off-policy RL algorithms. The experiments verify this on SAC and DDPG.

**Questions:**

Refer to above

**Limitations:**

Limitations are discussed in Section 7.

**Strengths And Weaknesses:**

**Strengths**

- The proposed method appears to be simple and effective empirically.
- The proposed method is evaluated in several configurations.
- The writing is clear and easy to follow.

**Weaknesses**

- The main concern is why the proposed method is not evaluated on DQN. The proposed method is claimed to be general ("The value target lower bounds can be readily plugged into RL algorithms that regress value to a target, e.g. DQN, DDPG or SAC."), but only evaluated on off-policy actor-critic style methods, like SAC and DDPG. Is this because it does not work well on DQN?
- Currently, the empirical return (or n-step return) is taken as the lower bounding function, which limits the proposed method to only valid when the environment is deterministic. This is one limitation of this work.
- *Faster learning* is only shown empirically and intuitively. Are there theoretical results to support this claim?
- It will be better to show the difference between the lower bounding and upper bounding (He et al., 2017).

**originality and significance**
- It is not totally new to use the bounding of target value.

---
AFTER REBUTTAL

I thank the authors for the response. After reading the response and other reviews, I think the paper still has a few weaknesses/limitations: (1) limited to deterministic environments. It is better to show how to extend it to stochastic environments; (2) better to compare it with other methods, like upper bounding.

Nevertheless, I still enjoy seeing such a simple method works well empirically. So, I keep my score.

---

> ### Author Response · Authors · 2022-07-30
> **Reply to Reviewer zNRT**
>
> Thanks for the careful review and detailed comments.
>
> 1. DDQN was evaluated and tends to be lower in performance than SAC, hence only reported in the Appendix 1.6.  Similar improvements were observed comparing value target lower bounding with DDQN.
>
> 2. Regarding limitation to deterministic environments: This work aims to present a simple, effective, efficient, carefree (no hyperparameter) and theoretically justified approach.  Although the example lower bounds are limited to deterministic environments, the theory is generally applicable to stochastic environments.  A similar prior work to compare would be Hindsight Experience Replay (HER), which is simple, effective, efficient, and also limited to deterministic environments [1].  However, unlike our work, HER relies on the task being goal conditioned with full knowledge of the reward function, has one hyperparameter to tune (the proportion of hindsight experience), and is not justified in theory for stochastic environments.  Our work shows further significant gains on top of HER on hard continuous control tasks.  (We included this discussion in the updated main paper.)
>
> 3. In theory, faster learning happens for certain iterations of training, where the distance to the optimal value is strictly smaller than $\gamma ||v - v^*||_{\infty}$.  See the proofs in Appendix 1.1.  We’ve also included a discussion of this in the appendix of the new draft, right after the proofs.
>
> 4. We’d like to leave upper bounding experiments and analyses to future work, given its limited scope and novelty but significantly more resources for tuning and experiments.
>
> [1] Léonard Blier and Yann Ollivier, Unbiased Methods for Multi-Goal Reinforcement Learning, CoRR 2021 https://arxiv.org/abs/2106.08863

---

### Official Review · Reviewer_41F1 · 2022-07-11

**Rating:** 3
**Confidence:** 3
**Soundness:** 2 fair
**Presentation:** 2 fair
**Contribution:** 2 fair

**Summary:**

This paper proposes value target lower bounding to improve the Bellman value target during value function learning. The proposed method computes and utilizes a lower bound for the value target during Bellman backup, and the resulting lower-bounded Bellman operator is claimed to converge to the same optimal value as using the vanilla Bellman operator in the tabular case. Additionally, the paper presents several ways of estimating the lower bound function in practice, such as using the episodic return in episodic tasks and n-step bootstrapped return in non-episodic tasks, and shows the effectiveness of value target lower bounding on several RL environments.

**Questions:**

Questions & Suggestions:


1. [Technical Part]


- 1.a. Please check Theorem 2.3. If $f$ is updated during training, please include the update rule/equation for $f$ in the main paper and in Algorithm 1.


- 1.b. Value iteration (VI) results do not directly generalize to RL settings and are not sufficient to rigorously justify performance in RL in my opinion. The authors may refer to Rmax [1] or randomized value function [2] for examples of rigorous analysis in the RL setting.


- 1.c. Exploration is crucial for efficient/faster RL. In the example of Section 4.3, it would be more interesting to discuss for instance how value target lower bounding may help the agent reach the target state faster (with fewer number of episodes of training). Propagating the value back is a less interesting problem once the target state has been reached.


2. [Clarity]


- 2.a. In the actual implementation, is the lower bound function $f(s)$ fixed? or being constantly updated? It seems that the observed episodic return is written into the buffer and read out during training as the lower bound. So, $f(s)$ is updated by the newly-observed episodic return? Please try to clarify this in the paper revision.


- 2.b. Is a larger value always a better Bellman target? Breakout in Figure 2 and 3 could be a good counter-example. In Figure 3, lb­-DR on Breakout shows consistently 25-30% larger value targets than the baseline, which is also the highest percentage among all the 20 environments. However, the performance of lb-DR is much worse than the baseline (and I believe Breakout is a deterministic environment where the proposed lower-bounding should apply based on the claims in the paper). Further investigation is needed to better understand when lower bounding the target is helpful.


[1] Brafman, Ronen I., and Moshe Tennenholtz. "R-max-a general polynomial time algorithm for near-optimal reinforcement learning." Journal of Machine Learning Research 3.Oct (2002): 213-231.

[2] Osband, Ian, et al. "Deep Exploration via Randomized Value Functions." J. Mach. Learn. Res. 20.124 (2019): 1-62.





**Limitations:**

The authors have mentioned the limitation of their work in Section 6, and I would encourage the authors to discuss a bit more on scenarios where the proposed lower-bounding method may fail, e.g., why the performance is poor on Breakout? The potential negative societal impact has not been discussed in the paper. As an illustrative example, what if the algorithm was used by a malicious user to build a robot? Targeting tasks with (sparse) rewards that lead to some harmful behaviors for a society? I believe such kind of scenario could be discussed. In addition, what can and should be done to prevent it from happening?

**Strengths And Weaknesses:**

Strengths:

- The related work is well cited and discussed in the paper, and it is clear how this work differs from previous contributions.

- The proposed method is simple, reproducible, and compatible with most state-of-the-art RL algorithms.


Weaknesses:
- There are several technical concerns in the paper. Specifically,

    - I believe the main theoretical result, Theorem 2.3, does not hold if the lower bound function $f$ is fixed (or static). In the proof, for any $s$, where $f(s) \geq v^*(s)$ and $f(s) \geq B(v)(s)$, we have
    $$ |B_f(v)(s) – v^*(s)| = |max(B(v)(s), f(s)) – v^*(s)| = |f(s) – v^*(s)| = f(s) – v^*(s) = Constant$$
    So, the distance to the optimal value would not shrink under the new lower-bounded Bellman operator. If $f$ is updated together with the value function during learning, please do make this clear and clearly show how $f$ is updated together with $v$ in the main paper. Based on the current presentation, Theorem 2.3 does not hold.


    - Results from value iteration (VI) do not directly generalize to RL settings where the transition dynamics and the reward function are unknown to the RL agent. This leads to a fundamental problem in RL that does not exist in VI, exploration vs. exploitation. However, the paper did not mention or address/discuss the exploration problem when transitioning from VI to RL, and thus the proposed method is not well justified in the RL setting.


    - In general, the episodic return (or Monte Carlo estimate of the return) is not guaranteed to be a lower bound of the optimal value (or the expected return following an optimal policy). Although the authors mentioned this in Section 6 (Related Works) as the limitation of their work, it is still concerning to me in terms of the soundness and the algorithmic design of the proposed method.


- The clarity of the paper could be further improved, and the paper needs to be better organized in my opinion. I encourage the authors to move key results and plots to the main paper. Putting all experimental figures in the appendix would jeopardize the readability of the paper.


Overall, the paper feels more like a work in progress to me, and certain technical details may need a bit rework and the paper could be better organized. Unfortunately, I don’t think the current version can be accepted and vote for rejection.

---

> ### Author Response · Authors · 2022-07-30
> **Reply to Reviewer 41F1**
>
> Thanks for the careful review and detailed comments.
>
> 1.a: Theorem 2.3 states that as long as $f$ lower bounds the max achievable value, i.e. $f \le \bar{G}^v(s)$, no matter whether $f$ is static or updated during training, value converges to the true value $v^*$.  Appendix 1.1 proves Theorem 2.3.  The counter example provided by the reviewer (a fixed $f$ where $f(s) \ge v^*(s)$ and $f(s) \ge B(v)(s)$) does not meet the condition of Theorem 2.3, because as training proceeds, the max achievable value $\bar{G}^v(s)$ converges to $v^*(s)$, and since $f$ is a lower bound of $\bar{G}^v(s)$, $f$ cannot be always above $v^*(s)$.
>
> 1.b: Thanks for pointing out the more general theoretical RL setting, example treatments, and the exploration problem.  We agree that theoretical results in the more general RL setting seem like an interesting direction to explore, and value target lower bounding might benefit exploration as well.  Currently, given the simplicity of the value target lower bounding approach and its effectiveness, we’d rather limit the scope of the theory to the tabular Q learning case, which appears non-trivial already.  Empirically, we rely on the baseline RL algorithms’ random policy for exploration, e.g. $\epsilon$-greedy for DDPG and entropy regularized policy for SAC.  New work [1] has also shown regret bounds for epsilon greedy algorithms, which adds a layer of safety to the current approach.
>
> 1.c: Agree, exploration is an important and interesting problem.  It can be a good topic for another research project, but maybe not for the current paper, given it’s already fairly long.
>
> 2.a: We use empirical return for trajectories from the replay buffer as the lower bound, which could change during the course of training.  Again, Theorem 2.3 applies as long as $f$ lower bounds the maximum achievable value, $f$ being static or not.  Line 93 of the draft mentions that “the lower bound doesn’t have to be static”.
>
> 2.b: Thanks for bringing up the point of failure analysis.  We did look into Atari Breakout.  The lower bounding method did achieve a much higher discounted return than the SAC baseline (without any apparent value overestimation), but it actually achieved a much lower evaluation score (total episodic reward).  This is likely because of the discrepancy between the training objective and the test metric:
>
>    During training, raw rewards are clipped to [-1, 1] and step discounted at $\gamma = 0.99$ to compute value, while in evaluation, total reward is the unclipped and undiscounted cumulative sum of episode rewards.  The discount and clipping together severely penalizes large rewards earned later in the episode, which is what’s happening for Breakout, because hitting a top layer block produces a reward of 7 while hitting a bottom layer block produces 1.  When we used non-clipped rewards or a higher $\gamma$ in training, the lower bounding method performed much better in total reward.  Note, this train-test discrepancy is likely present in all the prior works using policy gradient methods on Atari games.
>
>    We added this discussion to the updated main paper.
>
> Appendix 1.9 has been added to discuss ethical considerations/potential negative societal impact.
>
> [1] Chris Dann, Yishay Mansour, Mehryar Mohri, Ayush Sekhari, Karthik Sridharan, Guarantees for Epsilon-Greedy Reinforcement Learning with Function Approximation, ICML 2022.

---

> > ### Comment · Reviewer_41F1 · 2022-08-06
> > **Thank You**
> >
> > Thanks very much for the detailed response. Unfortunately, my major concerns for the paper still remain, and I will keep my original score and assessment.

---

### Meta-Review · Area_Chair_NRPe · 2022-08-23

**Recommendation:** Reject
**Confidence:** Certain

**Metareview:**

After reading the reviews and the author feedback, I lean towards rejection. The paper is not ready for publication for the following reasons:

 - Reviewers found that there is a lack of comparisons to other baselines and not enough discussion on the difference between lower and upper bounding.

- Reviewers found that the proposed algorithm has a limited scope (deterministic environments) and would like to know more about how to extend it to more general setting.

However, the clarity and the simplicity of the approach have been appreciated by the reviewers. Therefore, we encourage the authors to improve their paper by answering reviewers concerns and resubmit.

**Award:**

No

---

### Decision · Program_Chairs · 2022-09-14

Reject